# Validation-based model selection for $^{13}$C metabolic flux analysis with uncertain measurement errors

Nicolas Sundqvist[1], Nina Grankvist[2,3,4], Jeramie Watrous[5], Jain Mohit[5], Roland Nilsson[2,3,4], Gunnar Cedersund[1,6]*

1 Linköping's University, Department of Biomedical engineering, Linköping, Sweden, 2 Cardiovascular Medicine Unit, Department of Medicine (Solna), Karolinska Institutet, Stockholm, Sweden, 3 Division of Cardiovascular Medicine, Karolinska University Hospital, Stockholm, Sweden, 4 Center for Molecular Medicine, Karolinska Institutet, Stockholm, Sweden, 5 Department of Medicine & Pharmacology, University of California, San Diego, California, United States of America, 6 Center for Medical Image Science and Visualization, Linköping University, Linköping, Sweden

* gunnar.cedersund@liu.se

**Data Availability Statement:** All relevant data and supporting information are avilable at: https://gitlab.liu.se/nicsu70/val-based-model-selection_13cmfa.

## Abstract

Accurate measurements of metabolic fluxes in living cells are central to metabolism research and metabolic engineering. The gold standard method is model-based metabolic flux analysis (MFA), where fluxes are estimated indirectly from mass isotopomer data with the use of a mathematical model of the metabolic network. A critical step in MFA is model selection: choosing what compartments, metabolites, and reactions to include in the metabolic network model. Model selection is often done informally during the modelling process, based on the same data that is used for model fitting (estimation data). This can lead to either overly complex models (overfitting) or too simple ones (underfitting), in both cases resulting in poor flux estimates. Here, we propose a method for model selection based on independent validation data. We demonstrate in simulation studies that this method consistently chooses the correct model in a way that is independent on errors in measurement uncertainty. This independence is beneficial, since estimating the true magnitude of these errors can be difficult. In contrast, commonly used model selection methods based on the $\chi^2$-test choose different model structures depending on the believed measurement uncertainty; this can lead to errors in flux estimates, especially when the magnitude of the error is substantially off. We present a new approach for quantification of prediction uncertainty of mass isotopomer distributions in other labelling experiments, to check for problems with too much or too little novelty in the validation data. Finally, in an isotope tracing study on human mammary epithelial cells, the validation-based model selection method identified pyruvate carboxylase as a key model component. Our results argue that validation-based model selection should be an integral part of MFA model development.

**Funding:** This work was supported by the Swedish Foundation for Strategic Research, grant no. FFL12-0220 (RN, NG) and IMT17-0245 (RN, NG, NS, GC), the Swedish Research Council grant no. 2018-05418 (GC) and 2018-03319 (GC, NS), Karolinska Institutet (RN), CENIIT grant no. 15.09 (GC), SciLifeLab National COVID-19 Research Program, financed by the Knut and Alice Wallenberg Foundation grant no. 2020.0182 (GC), the H2020 project PRECISE4Q grant no. 777107 (GC), and from VINNOVA grants VisualSweden and 2020-04711 (GC). Additional support for GC came from the Swedish Fund for Research without Animal Experiments (F2019-0010), ELLIIT (2020-A12). Initials indicate that part of the author's salary has been paid for by the preceding funds. The funders had no role in study design, data collection and analysis, decision to publish, or preparation of the manuscript.

**Competing interests:** The authors have declared that no competing interests exist.

## Author summary

Measuring metabolic reaction fluxes in living cells is difficult, yet important. The gold standard is to label extracellular metabolites with $^{13}$C, to use mass spectrometry to find out where the $^{13}$C-atoms ends up, and finally use mathematical modelling to calculate how quickly each reaction must have flowed, for the $^{13}$C-atoms to end up like that. This measurement thus relies on usage of the right mathematical model, which must be *selected* among various candidate models. In this manuscript, we present a new way to do this model selection step, utilizing validation data. Using an adopted approach to calculate the uncertainty of model predictions, we identify new validation experiments, which are neither too similar, nor too dissimilar, compared to the previous training data. The model candidate that is best at predicting this new validation data is the one chosen. Tests on simulated data where the true model is known, shows that the validation-based method is robust when the magnitude of the error in the measurement uncertainty is unknown, something that conventional methods are not. This improvement is important since true uncertainties can be difficult to estimate for these data. Finally, we demonstrate how the new method can be used on real data, to identify fluxes and important reactions.

## 1. Introduction

Cellular metabolism is fundamental for all living organisms, involving thousands of metabolites and metabolic reactions that together form large interconnected metabolic networks [1,2]. While a substantial part of the human metabolic network has been reconstructed [2], measuring fluxes through individual reactions and metabolic pathways in living cells and tissues remains a challenge. This problem is central to a variety of medically relevant processes, including T-cell differentiation [3], caloric restriction and aging [4], cancer [5,6], the metabolic syndrome [7], and neurodegenerative diseases such as Parkinson's disease [8].

The gold standard method for measuring metabolic fluxes in a given system is model-based metabolic flux analysis (MFA) [9]. In this technique, cells or tissues are fed "labelled" substrates containing stable isotopes such as $^{13}$C (Fig 1A). These substrates are metabolized to products containing various isotopic isomers (isotopomers) (Fig 1B). By measuring the abundance of these isotopomers, mass isotopomer distributions (MIDs, Fig 1C) are obtained for each metabolite [10]. Fluxes are then inferred by fitting a mathematical model $\mathcal{M}$ to the observed MID data $D$ (Fig 1D).

While the above methodology is well established for assessing the fit of a given MFA model, several problems arise when it is used for model selection. In practise, MFA models are usually developed iteratively (Fig 1D), by repeatedly attempting to fit the same data to a sequence of models $\mathcal{M}_1, \mathcal{M}_2, \ldots, \mathcal{M}_k$ with successive modifications (adding or removing reactions, metabolites, and so on), until a model $\mathcal{M}_k$ is found acceptable, i.e. not statistically rejected. In practice, this means that the model $\mathcal{M}_k$ passes the $\chi^2$-test for goodness-of-fit [11]. Given the iterative nature of modifying the model structures, model development thus turns into a model selection problem. Depending on the approach used to solve this model selection problem different model structures might be selected, given the same data set (Fig 1E). For instance, if the traditionally iterative modelling cycle is used, the first model that passes the $\chi^2$-test might be selected and used for flux estimation. On the other hand, there might be multiple model structures that passes the $\chi^2$-test. In this case, the model structure that passes the $\chi^2$-test with the biggest margin may be a better option.

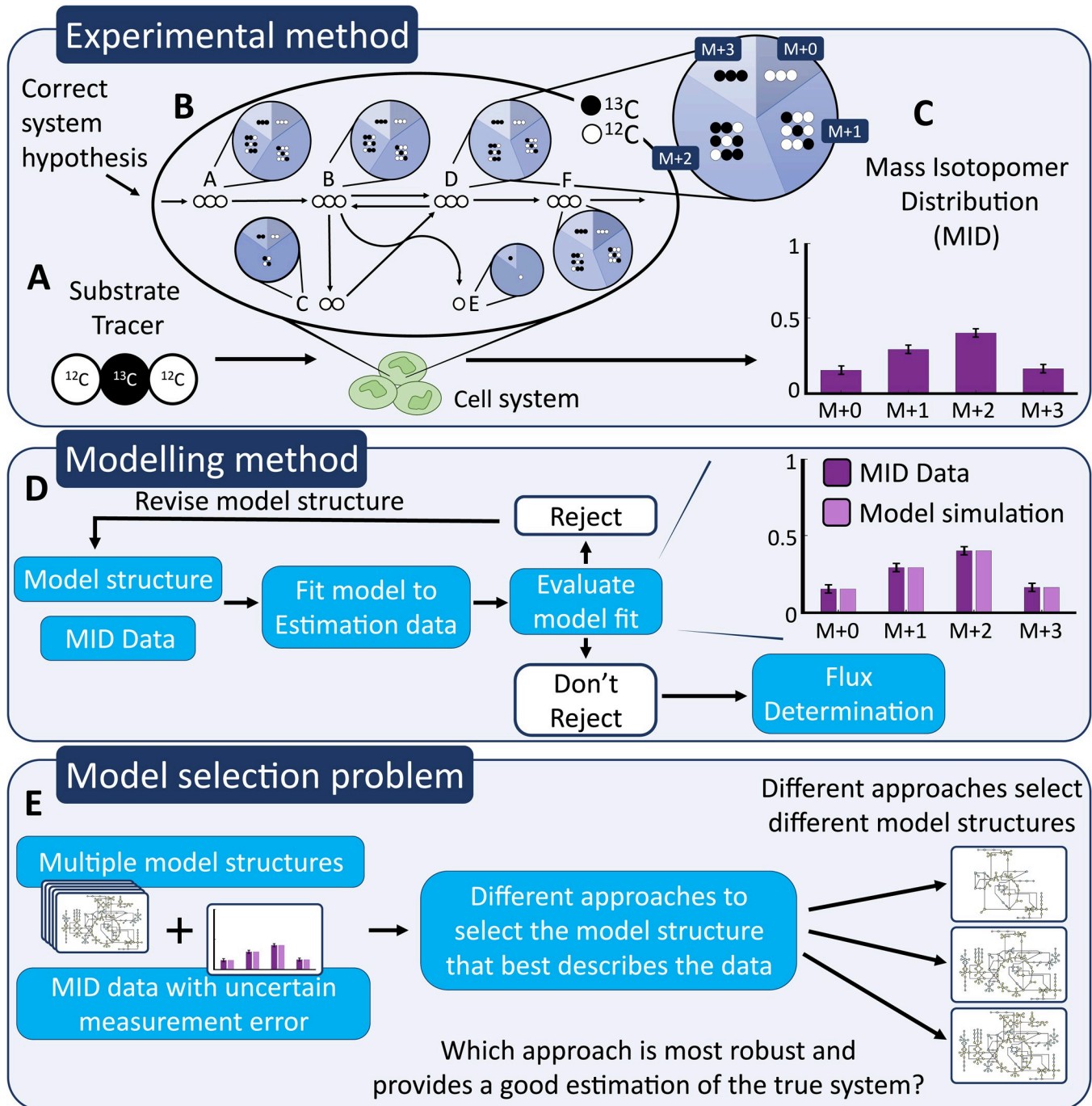

**Fig 1. The basic steps in $^{13}$C MFA and the model selection problem.** (A) New substrates, containing $^{13}$C (dark circles) are fed to the cells. (B) These substrates are consumed and converted to end products in the cells, according to its biochemical reactions. (C) The labelled $^{13}$C molecules appear to various proportions in each of the mass isotopomers, and these proportions are summed up in these distribution bar charts for each detected metabolite. (D) The iterative modelling cycle in which a hypothesized model structure is fitted to MID data. The model fit is evaluated, usually with a $\chi^2$-test, and either rejected or not. If the model structure is rejected it is revised and evaluated again. If the model structure is not rejected it is used for flux determination. (E) The iterative model development in (D) results in a model selection problem. Different approaches for solving this model selection problem might result in different model structures being selected. This paper evaluates how the uncertainty in measurement data affects uncertainty in model selection.

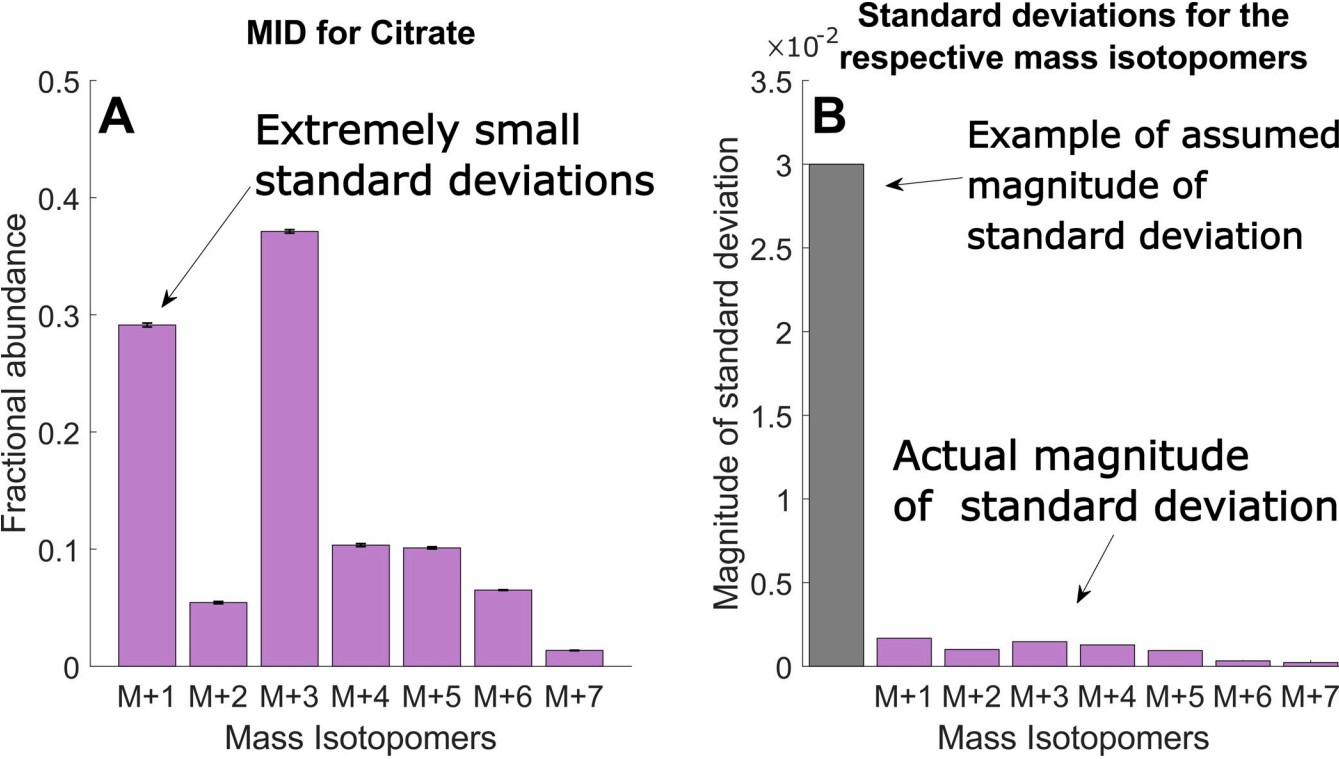

**Fig 2. Example of MID sample standard deviation (A) Example of estimated mass isotopomer distribution (MID) of citrate from epithelial cells, as described in section 2.5.** M+i indicate the fractional abundance of the i:th mass isotopomer. (B) Difference between the assumed magnitude of the standard deviations and the measured magnitudes.

Generally, model selection approaches that rely solely on the $\chi^2$-test to select a model can be problematic. First, correctness of the $\chi^2$-test depends on knowing the number of identifiable parameters, which is needed to properly account for overfitting by adjusting the degrees of freedom of the $\chi^2$ distribution [12], but can be difficult to determine for nonlinear models [13]. Second, the $\chi^2$-test can be unreliable in practise since the underlying error model is often not accurate. Typically, the MID errors $\sigma$ are estimated by sample standard deviations $s$ from biological replicates, which for mass spectrometry data often is below 0.01, and even can be as low as 0.001 (Fig 2A). However, such low estimates may not reflect all error sources. For example, MI fractions obtained from orbitrap instruments can be biased so that minor isotopomers are underestimated [14,15]. Also, $s$ does not account for experimental bias, such as deviations from metabolic steady-state that always occur in batch cultures. Some such problems can be detected by repeating experiments, but some others cannot. The normal distribution assumption itself is also questionable for MIDs, which are constrained to the n-simplex [16]. For these reasons, $s$ can severely underestimate the actual errors, making it exceedingly difficult to find a model that passes a $\chi^2$-test. In this situation, one is left with two bad choices: either arbitrarily increase $s$ to some "reasonable" value to pass the $\chi^2$-test (Fig 2B), or introduce more or less well-motivated extra fluxes into the model. The former alternative, increasing $s$, may lead to high uncertainty in the estimated fluxes and does not necessarily reflect the experimental bias one tries to account for. The latter approach, introducing additional fluxes, increases model complexity and can lead to overfitting.

While these issues with model selection are well known, they have to our knowledge not been treated systematically in the $^{13}$C MFA field. Indeed, MFA model selection is typically

done in an informal fashion by trial-and-error, and the underlying procedure is rarely reported [17]. However, in other contexts where model fitting is central, such as systems biology, the problem of model selection has been treated extensively [13,18–27]. In these areas, a widely accepted solution is to perform model selection on a separate "validation" data set, which is not used for model fitting. Intuitively, this protects against overfitting by choosing the model that can best predict new, independent data. In this paper, we propose a formalized version of such a validation-based model selection approach for MFA. In a series of simulated examples, we demonstrate that this method consistently selects the correct metabolic network model, despite uncertainty in measurement errors, whereas "traditional" $\chi^2$-testing on the estimation data does not. By quantifying prediction uncertainty using prediction profile likelihood, we can avoid cases where the validation data is too similar, or too dissimilar, to the estimation data. Finally, in an application to flux analysis on our own new data in human epithelial cells, we find that the same robustness to measurement uncertainty variations still holds, and that the validation-based model selection method can identify reactions that are known to be active in this cell type.

## 2. Results

To systematically examine the effects of the model selection procedure on MFA, we adopted a scheme where a sequence of models $\mathcal{M}_1, \mathcal{M}_2, \ldots$ with increasing complexity (increasing number of parameters) is tested by each model selection method, simulating typical iterative model development. We considered five possible model selection methods that use all available data for both parameter estimation and model evaluation (Table 1). Method "SSR" selects the model with the smallest weighted summed squared residuals (*SSR*) based on the data, included as a baseline. Method "First $\chi^2$" selects the model with fewest parameters (the "simplest" model) that passes a $\chi^2$-test, while accounting for overfitting by subtracting the number of free parameters *p* from the degrees of freedom in the $\chi^2$-distribution (see Section 4.3). Method "Best $\chi^2$" selects the model that passes the $\chi^2$-threshold with the greatest margin. Methods "AIC" and "BIC" select the model that minimizes the Akaike Information Criterion or the Bayesian Information Criterion, respectively [28,29]. The five methods mentioned above all depend on the noise model Eq (5), and all except "SSR" also requires knowing the number of free parameters *p*. Considering common practices in the field, it is probable that some combination of the "First $\chi^2$" and "Best $\chi^2$" methods is the prevailing approach in MFA modelling [17,30], although this is not entirely clear since the model selection process is often not described.

In addition to these methods, we propose a validation-based model selection method ("Validation") that divides the data $D$ into estimation data $D^{est}$ and validation data $D^{val}$. For each model, parameter estimation (model fitting) is then done using $D^{est}$, and the model achieving the smallest SSR with respect to $D^{val}$ is selected. The division into estimation and validation

**Table 1. A summary of the different model selection approaches considered in this paper.**

| Method of model selection | Model selection criteria |
|---|---|
| Estimation SSR | Selects the model with the lowest SSR given $D^{est}$ |
| First $\chi^2$ | Selects the first $\mathcal{M}_k$ that passes the $\chi^2$-test |
| Best $\chi^2$ | Selects the $\mathcal{M}_k$ that passes the $\chi^2$-test with the greatest margin |
| AIC | Selects the $\mathcal{M}_k$ that minimizes the Akaike Information Criterion |
| BIC | Selects the $\mathcal{M}_k$ that minimizes the Bayesian Information Criterion |
| Validation | Selects the $\mathcal{M}_k$ with the smallest SSR with respect to $D^{val}$ |

data must be done so that qualitatively new information is present in the validation data. This can be done by reserving data from distinct model inputs or new model outputs for validation. For all examples herein, data from distinct model inputs is used for validation. For the $^{13}$C MFA examples, this means that data used for validation comes from a different tracer. Note that this proposed method allows for the selection of the most suitable model from a given set, but that it does not guarantee that the selected model is acceptable according to e.g. a $\chi^2$-test. In other words, the model selected with our new Validation method would still need be subjected to some form of final model testing. A detailed description of the "SSR", "First $\chi^2$", "Best $\chi^2$", and "Validation" methods can be found in S1 Algorithm: A-D respectively (S1 Algorithm).

## 2.1 A motivating example

Before examining the behavior of the different model selection methods on metabolic network models, it may be helpful to illustrate their properties on a simple univariate example. For this purpose, we considered a model with a single input $x$ and a single output $\hat{y}$, where model $\mathcal{M}_n$ is the n-th order polynomial

$$\hat{y} = h_n(x, u) = \sum_{k=0}^{n} u_k x^k \tag{1}$$

with parameter vector $u$. We assume that $\mathcal{M}_7$ is the correct model, with true parameters $u_0$, and sampled 20 measurements $y = h_7(x, u_0) + \epsilon$ for different values of $x$, where $\epsilon$ was drawn from $N(0, \sigma_r)$ with standard deviation $\sigma_r = 0.2$. To simulate uncertainty about the error model, we considered $\sigma$ to be unknown, and let the various model selection methods choose among $\mathcal{M}_1, \ldots, \mathcal{M}_{14}$ with a "believed" standard deviation, denoted $\sigma_b$, in the range [0.1 $\sigma_r$, 10 $\sigma_r$]. For the "Validation" method, we reserved 4 of the 20 measurements for $D^{val}$ (Fig 3, red error bars).

An illustration of the dependency on $\sigma$ for a model selection method that does not use validation is shown in Fig 3. When $D^{val}$ is not considered, we would expect larger values of $\sigma_b$ to result in a simpler model, since almost all of the variation in the data is interpreted as noise (Fig 3A). Further, at very small values of $\sigma_b$ an overly complex model will be required to obtain an acceptable fit to $D^{est}$ (Fig 3C).

Applying the five model selection methods to data from this polynomial model gave different results (Figs 4 and S1). Since the model selection process is somewhat stochastic, we resampled the data 10,000 times, each time with a new error $\epsilon$ drawn from $N(0, \sigma_r)$, and report results as the fraction of times a particular model was chosen. As expected, "SSR" mostly selected the most complex polynomial $\mathcal{M}_{14}$ regardless of $\sigma_b$, as the most complex model always gives the lowest SSR (Fig 4A). In contrast, "First $\chi^2$" or "Best $\chi^2$" gave different results depending on $\sigma_b$. "First $\chi^2$" selected the correct model $\mathcal{M}_7$ only when $\sigma_b \approx \sigma_r$. At $\sigma_b \approx 10\sigma_r$, only the low-degree polynomials ($\mathcal{M}_1, \mathcal{M}_2$ and $\mathcal{M}_3$) was chosen by the "First $\chi^2$" method, while at $\sigma_b \approx 0.1\sigma$, an overly complex polynomial was chosen (Fig 4B). The "Best $\chi^2$" method selected the correct model $\mathcal{M}_7$ for $\sigma_b \geq \sigma_r$, but selected overly complex models for smaller $\sigma_b$ (Fig 4C). If $\sigma_b$ were to increase further, "Best $\chi^2$" would choose a lower degree polynomial. This is because, for these $\chi^2$-based methods, the tradeoff between model complexity and goodness-of-fit is based on $\sigma_b$, and such a tradeoff is thus correct only if we happen to have $\sigma_b \approx \sigma_r$. Similar results are seen with the "AIC" and "BIC" methods, which also depend on $\sigma_b$ (S1 Fig).

In contrast, the "Validation" method predominantly selected the correct model, $\mathcal{M}_7$, regardless of $\sigma_b$ (Fig 4D). This happens because, even though a polynomial of the wrong degree may fit well on $D^{est}$, it fails to predict independent validation data, resulting in large SSR on $D^{val}$. Since the correct model structure $\mathcal{M}_7$ will best predict new data, agreement with validation data helps identify the right model, also in cases where the error model is inaccurate.

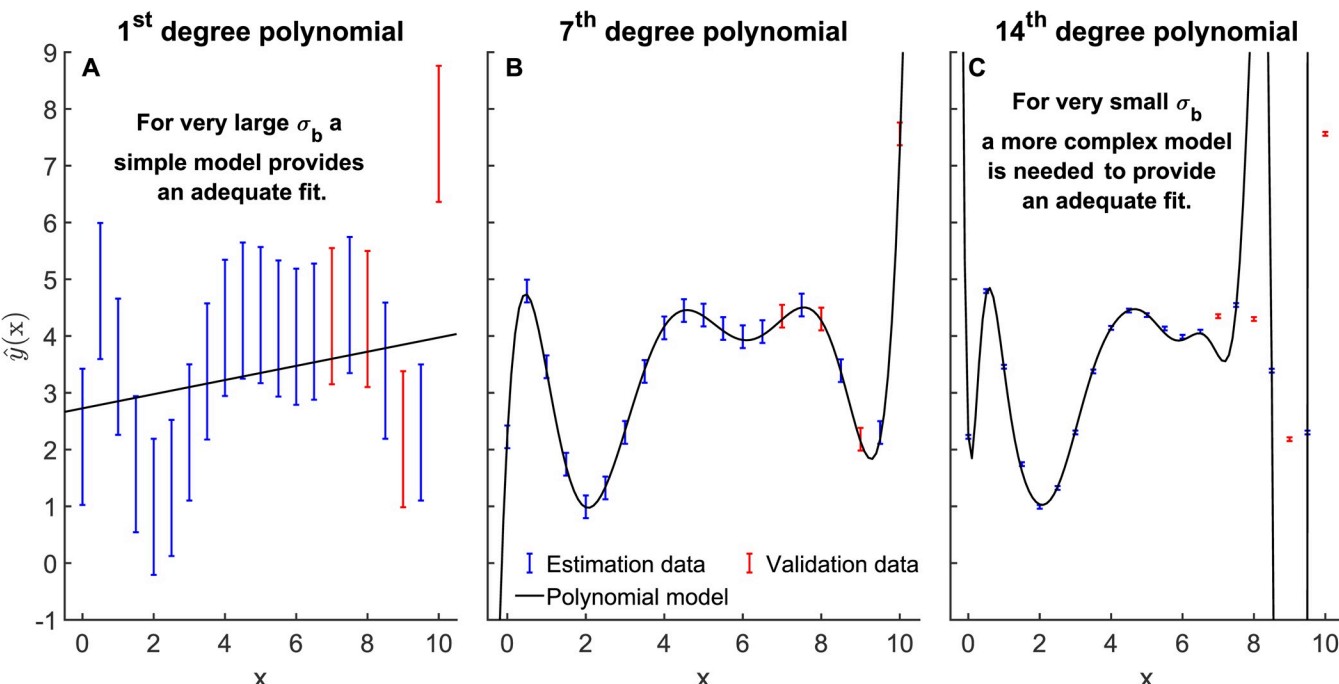

**Fig 3.** Example of how model selection is affected by $\sigma_b$, for the polynomial model. Error bars indicate data sampled from a 7$^{th}$ order polynomial $y = h_7(x, u_0)$ $+ \epsilon$ where $\epsilon$ is N(0, $\sigma_r$), $\sigma_r = 0.2$. Colours indicate estimation data D$^{est}$ (blue) and validation data D$^{val}$ (red) used by the "Validation" method. Solid curves in (A–B) indicate polynomials chosen by an estimation-based method with different "believed" standard deviation $\sigma_b$. (A) $\sigma_b = 2$, chosen model $h_1$. (B) $\sigma_b = 0.2$ (the true value), chosen model $h_7$ (the correct model). (C) $\sigma_b = 0.02$, chosen model $h_{14}$.

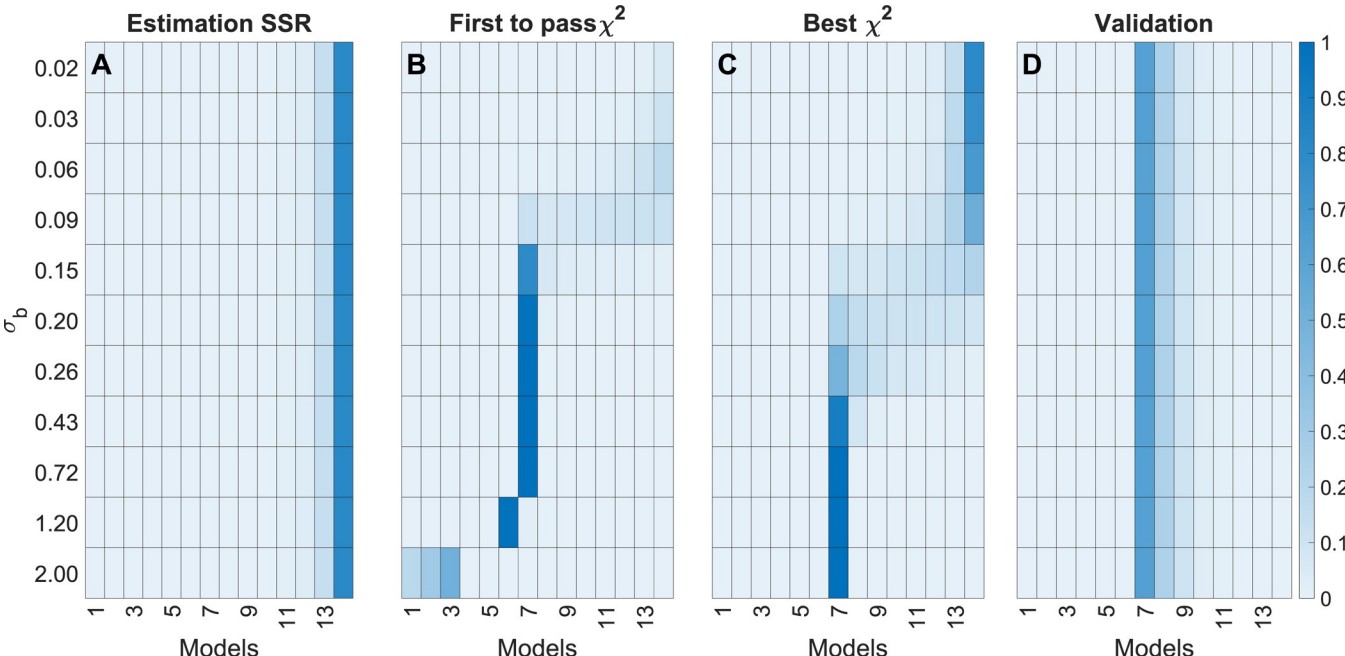

**Fig 4. Model selection results for the polynomial model example.** (A–D) Heatmaps represent results from the indicated selection methods, where rows represent different values of $\sigma_b$ and columns represent the polynomial models $h_1, \ldots, h_{14}$. For each row, color indicates the fraction of times a model is selected for the given $\sigma_b$, out of 10,000 samples, as indicated by the color scale (right).

Again, it should be recalled that the "Validation" method only is applicable to the task of selecting the best model, and that this selection approach should be followed by a final step that tests the quality of the model, e.g. using a $\chi^2$-test.

## 2.2 Model selection on multivariate linear models

To investigate model selection in a setting more relevant to metabolic networks, we next considered a multivariate linear model, where the output vector $\hat{y}$ is a linear combination of the inputs $x$ weighted by the model parameters. In this case, each model structure $\mathcal{M}_k$ is fully specified by a matrix $A_k$ such that

$$\hat{y} = h_k(x, u) = A_k x \qquad (2)$$

and where the free parameters are elements of $A_k$ (Fig 5). This type of model is roughly analogous to a simple metabolic network, where $x$ corresponds to labelled substrates and $y$ corresponds to metabolic products. We constructed six such models ($A_1$–$A_6$) of increasing complexity, nested so that the parameter space of each $A_k$ contains the parameter space of all models $A_l$ for $l<k$. Model $A_3$ was used to simulate data from 6 distinct input vectors $x$, again with normal noise $N(0, \sigma_r)$ where $\sigma_r = 5$, and with a "believed" $\sigma_b$ in the range [0.1 $\sigma_r$, 10 $\sigma_r$]. The believed $\sigma_b$ is not scaled homogeneously across all data points rather the scaling is approximately $0.1\sigma_r$ and $10 \sigma_r$, to reflect the more realistic scenario of $\sigma_b$ being wrong to different degrees for different data points (Materials and methods). Again, to account for variance in the model selection process, the results are based on 1,000 different resamplings.

We then tested each of the six model selection methods on the generated data. As before, the "SSR" method chose the most complex models ($A_5$ and $A_6$, Fig 6A). For the other four methods that only use estimation data, the selected model again depended on $\sigma_b$. Method "First $\chi^2$" selected the one of the simpler models $A_2$ at $\sigma_b \approx 10 \sigma_r$, the correct model $A_3$ only when $\sigma_b \approx \sigma$, while at $\sigma_b \approx 0.1 \sigma_r$, no model passed the $\chi^2$-test (Fig 6B). The "Best $\chi^2$" method again selected the correct model, $A_3$, for $\sigma_b \approx 10 \sigma_r$ and $\sigma_b \approx \sigma_r$, and model $A_6$ for $\sigma_b \approx 0.1 \sigma_r$

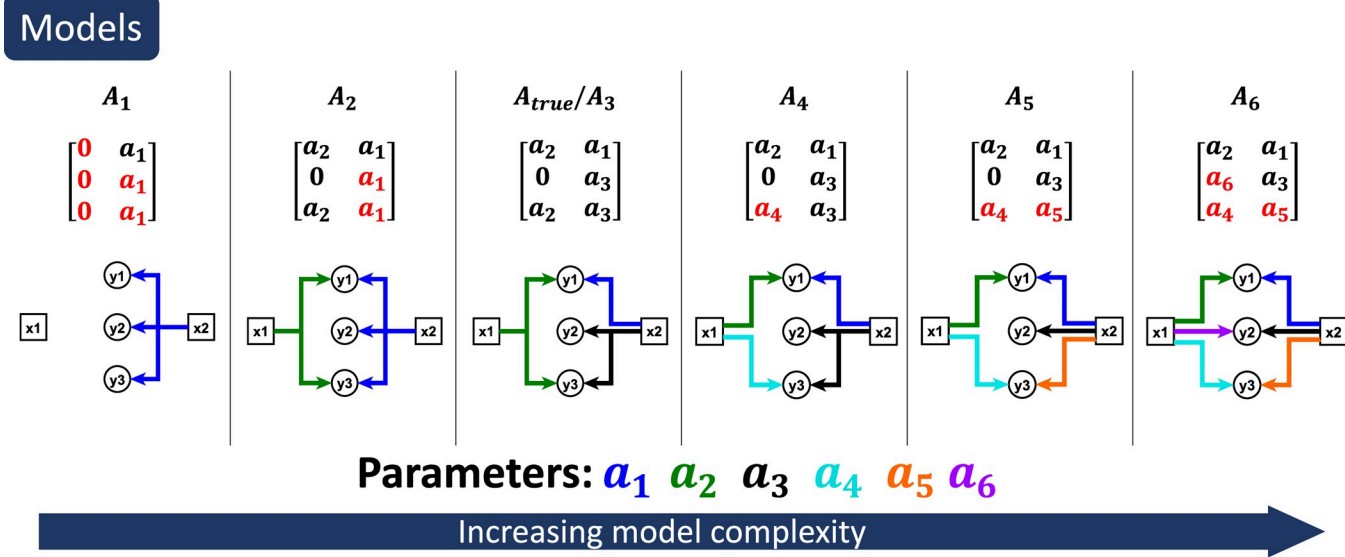

**Fig 5. Six different model structures for the linear model.** This example is chosen as a simple representation of a mass flow model. The top row shows the model names $A_1, \ldots, A_6$. The second row shows the matrices that constitute the model structures. The third row constitute visual illustrations of how the corresponding matrices connect the inputs $x_i$ and the outputs $y_i$ via the parameters $a_1, \ldots, a_6$.

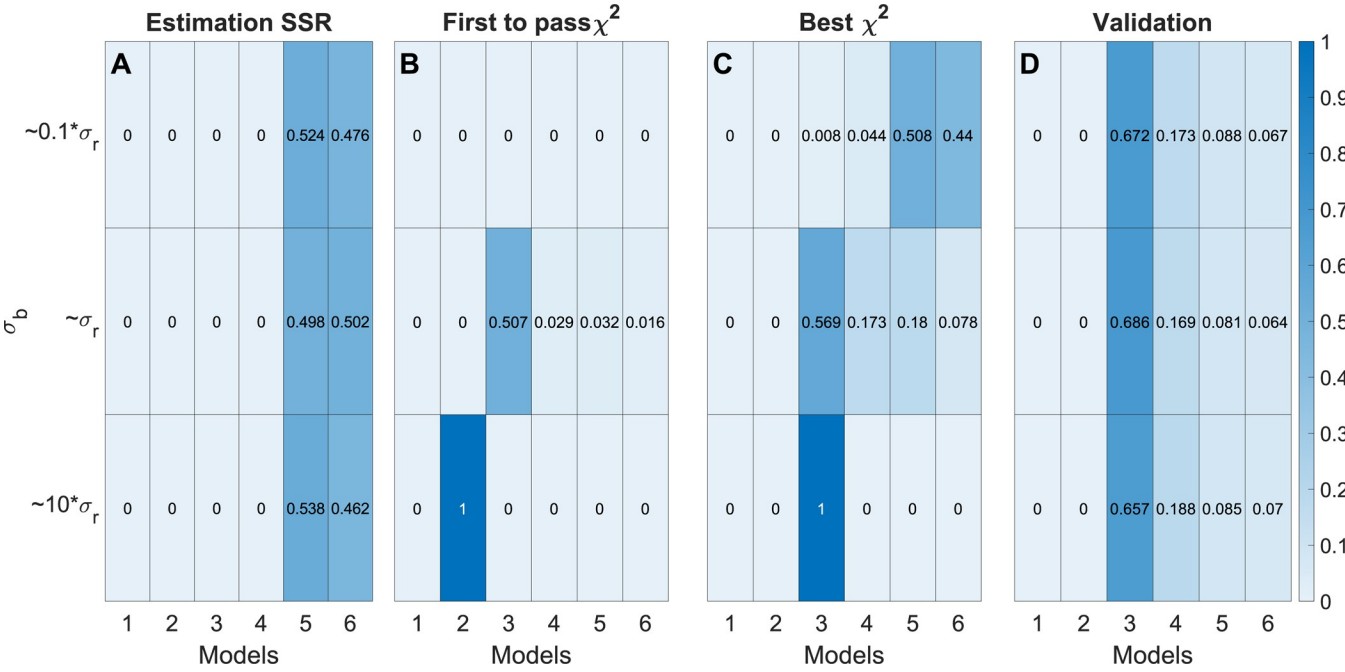

**Fig 6. Model selection results for the linear model example.** (A–D) Heatmaps represent results from the indicated selection methods, where rows represent different values of $\sigma_b$ and columns represent the linear models $A_1, \ldots, A_6$. For each row, color indicates the fraction of times a model is selected for the given $\sigma_b$, out of 1000 samples, as indicated by the color scale (right).

(Fig 6C). Methods "AIC" and "BIC" behaved similarly but chose somewhat different models (S2 Fig). Thus, which model is considered "best" depends on assumptions about the measurement noise, and established model selection methods give different results depending on what assumption is made.

For the "Validation" method, simulated data from 2 of the 6 distinct inputs x were reserved for validation data $D^{val}$ (S3 Fig). Again, this method predominantly selected the true model structure $A_3$, regardless of $\sigma_b$ (Fig 6D). Moreover, the model selection results for "Validation" method are consistent across all $\sigma_b$.

## 2.3 Model selection for simulated $^{13}$C MFA models

Let us now turn to model selection for the multivariate, nonlinear MFA models. To simulate the process of MFA model development, we designed seven stoichiometric models $\mathcal{M}_1, \ldots, \mathcal{M}_7$ of the tricarboxylic acid (TCA) cycle and related reactions, with increasing model complexity (S1 Table and Fig 7). The full, atom-level models were generated using the EMU decomposition method (see methods). For all seven models, 51 MI fractions across nine metabolites (present in all models) were considered as measurement data. The data was simulated (Section 4.5) using model $\mathcal{M}_4$, with four different tracers separately used as inputs x in order to generate four separate sets of MID data. For this example, we resampled the data 100 times. Note that, unlike the previous examples, this model is nonlinear in the parameters.

As in the previous examples, the six methods of model selection were evaluated. As before, the "SSR" method always selected the most complex model $\mathcal{M}_7$ (Fig 8A). Method "First $\chi^2$" selected different models depending on the value of $\sigma_b$ (Fig 8B): it selects $\mathcal{M}_1$ for $\sigma_b \approx 10\sigma_r$, $\mathcal{M}_2$ for $\sigma_b \approx 3\sigma_r$, $\mathcal{M}_4$ (the true model) and $\mathcal{M}_2$ about 50% of the time respectively for $\sigma_b \approx \sigma_r$, and $\mathcal{M}_4$ for $\sigma_b \approx 0.1\sigma_r$ and $0.3\sigma_r$. In this example, "Best $\chi^2$" method selected the correct model $\mathcal{M}_4$

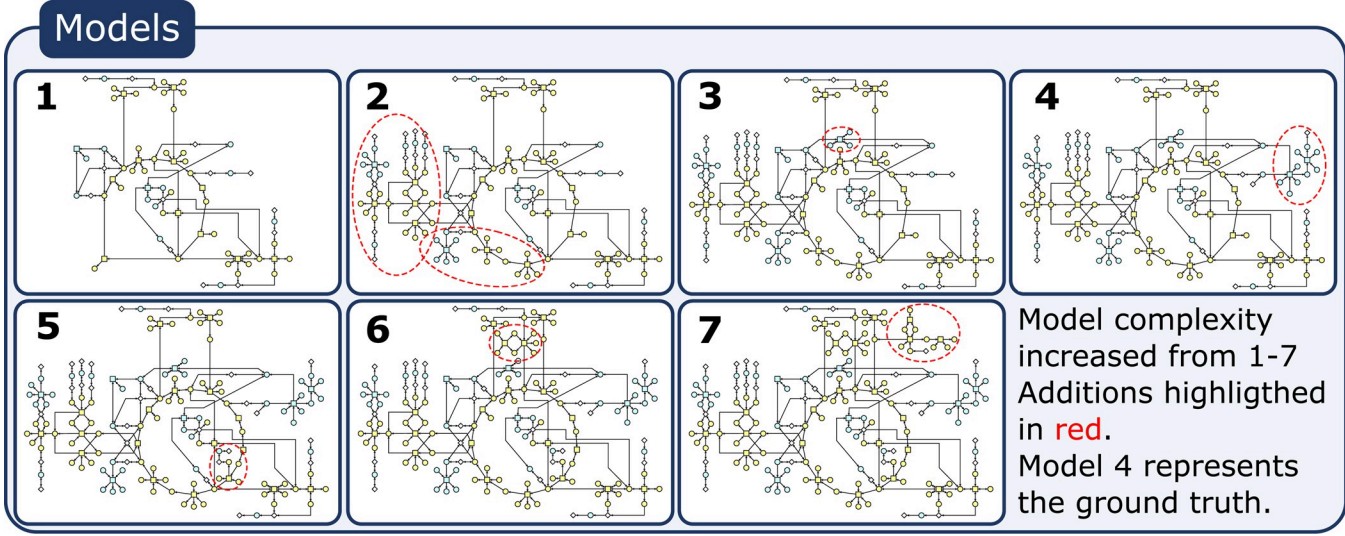

**Fig 7. Seven different model structures included in the simulated EMU $^{13}$C MFA example with simulated data.** The added component to each model structure, compared to the previous model, with slightly smaller complexity, is found inside the red circle. The true model used to simulate the data is model nr 4. Detailed descriptions for each model can be found in the supplementary material (S1 Table).

for $\sigma_b \approx 0.3\sigma_r$ and $\sigma_r$ (Fig 8C). For $\sigma_b \approx \sigma_r$ "Best $\chi^2$" show a fraction of the samples selecting $\mathcal{M}_2$ rather than $\mathcal{M}_4$ (Fig 8C) and for $\sigma_b \approx 10\sigma_r$, "Best $\chi^2$" shift towards selecting the simpler model structures $\mathcal{M}_2$. Compared to previous examples, the AIC and BIC methods of model selection appear to be a bit more robust towards an unknown $\sigma_b$, selecting $\mathcal{M}_4$ for $\sigma_b \approx 0.1\sigma_r$, $0.3\sigma_r$, $\sigma_r$, and $3\sigma_r$. Nevertheless, for $\sigma_b \approx 0.1\sigma_r$, both the AIC and BIC show tendencies to prefer more

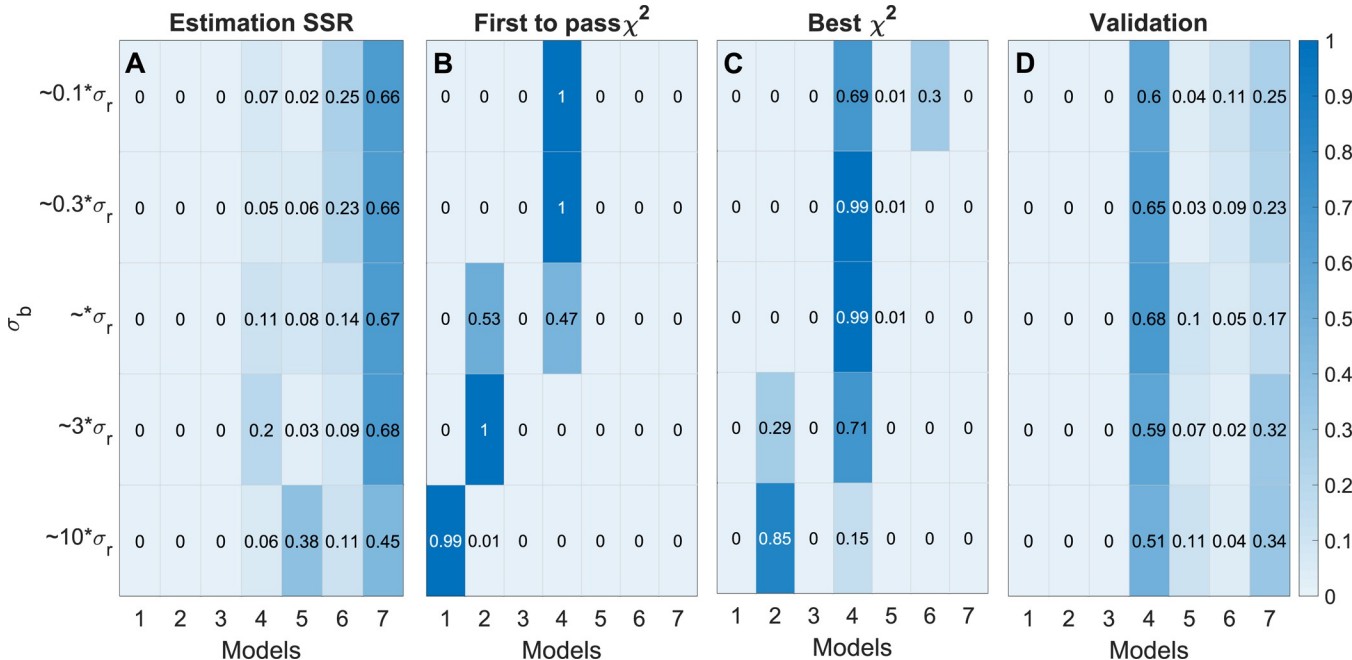

**Fig 8. Model selection results for the simulated $^{13}$C MFA model example.** (A–D) Heatmaps represent results from the indicated selection methods, where rows represent different values of $\sigma_b$ and columns represent the MFA models $\mathcal{M}_1, \ldots, \mathcal{M}_7$. For each row, color indicates the fraction of times a model is selected for the given $\sigma_b$, out of 100 samples, as indicated by the color scale (right).

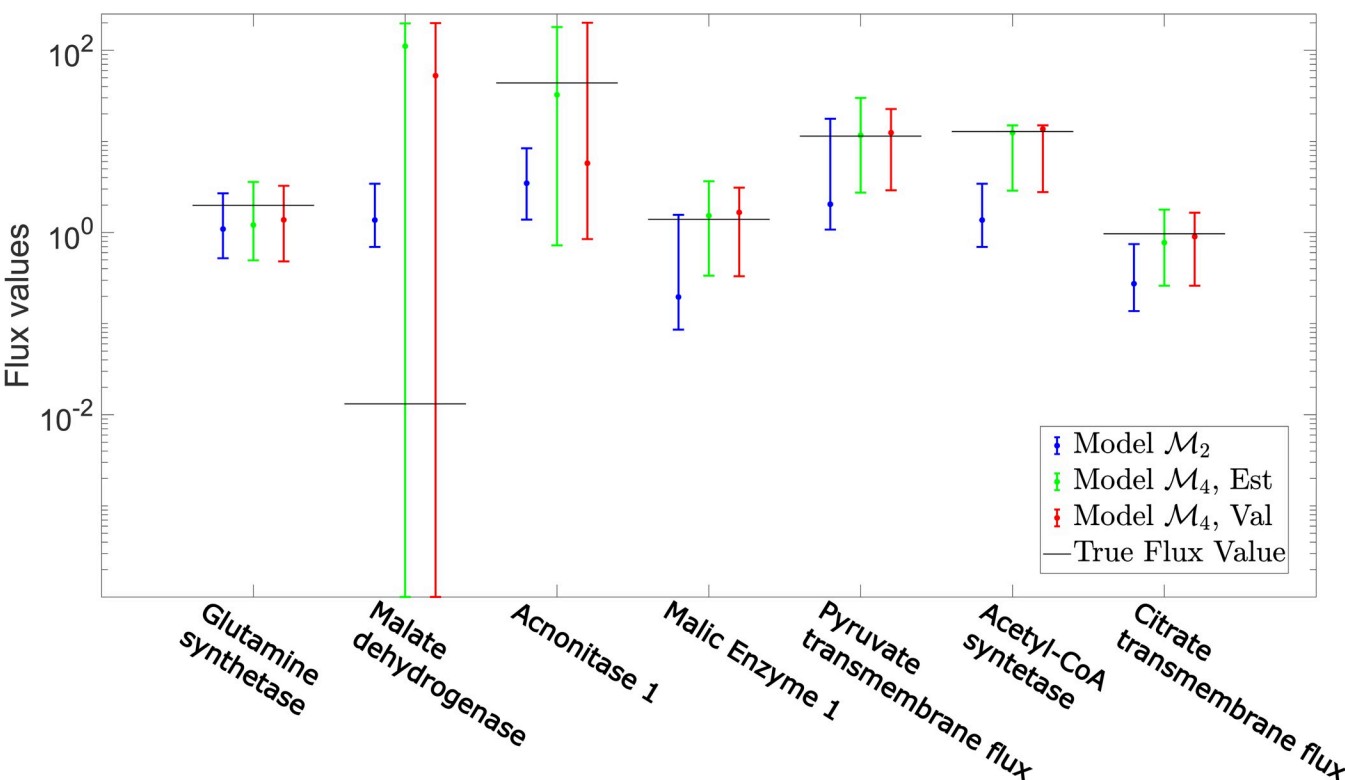

**Fig 9. Comparison of estimated flux solutions for the simulated $^{13}$C MFA example.** The resulting flux values with 95% confidence intervals for seven of the fluxes that are overlapping between all model structures in the simulated $^{13}$C MFA example. The confidence intervals correspond to the estimated fluxes for model $\mathcal{M}_2$ (Blue), model $\mathcal{M}_4$ with all data available (Green) and model $\mathcal{M}_4$ with the data split into D$^{est}$ and D$^{val}$ (Red). The figure illustrates the selecting the wrong model structure may result in incorrect flux estimations.

complex models and for $\sigma_b \approx 10\sigma_r$ both the AIC and BIC selects too simple model structures, namely $\mathcal{M}_2$ and $\mathcal{M}_1$ respectively (S4 Fig).

For the "Validation" method, parameters were estimated using MID data from 3 out of 4 tracers, while the fourth set of MID data was used as validation data $D^{val}$; the exact division is described in Section 4.5. The "Validation" method selected the correct model $\mathcal{M}_4$ in 60–70% of cases, for all tested values of $\sigma_b$ (Fig 8D). The key observation here is that the validation-based method obtains the same results independently of $\sigma_b$. However, it should be noted that the "Best $\chi^2$" method does appear more robust in identifying the correct model when $\sigma$ is correct.

By selecting the wrong model structure, methods that depend on $\sigma_b$ can lead to poor estimates of metabolic fluxes. For instance, when investigating the estimated flux values for model $\mathcal{M}_2$ (model selected by "*First $\chi^2$*" at $\sigma_b \approx 3\sigma_r$) it becomes clear that the "*First $\chi^2$*" approach does not always capture the correct flux value with a 95% confidence interval (Fig 9). For instance, for the fluxes for mitochondrial Aconitase1 (ACONT1m) and Acetyl-CoA Synthetase (ASCm), the 95% confidence interval lies several standard deviations away from the estimated value, indicating that the confidence intervals are not reliable. In contrast, the flux solution, with a 95% confidence interval, for model $\mathcal{M}_4$ (selected by the "Validation" method) does contain the true flux value (Fig 9). These results show that selecting the wrong model structure leads to errors in flux estimation, and that the "Validation" method therefore is more advantageous for both of these tasks.

## 2.4 Assessing the novelty of a validation experiment using prediction uncertainty

As showed in the previous examples, validation data can be used for the purpose of model selection. However, one important aspect to consider for this new method in $^{13}$C MFA, is the degree of novelty of the validation data used. There are essentially two pitfalls that one wants to avoid: 1) the validation data may be too similar to the estimation data (i.e. then the new data does not provide any new information), 2) the validation data may be too dissimilar from the estimation data (i.e. then no model is able to predict the validation data). Both of these pitfalls can be avoided by looking at the uncertainty of the model predictions for the chosen validation experiment. The model's prediction uncertainty is essentially a confidence interval for the predictions. In our case of $^{13}$C MFA models, a confidence interval is an interval for the predicted MIDs. This prediction uncertainty will depend on the uncertainty of the estimated fluxes, the model structure, and on the connection between the validation data and the estimation data. If the validation data is not novel enough (pitfall 1 above), models will produce identical predictions which do not differ from the estimation data (Fig 10A). On the other hand, if the validation data is too novel (pitfall 2), the estimation data does not contain any new information regarding the predicted MIDs, and the uncertainty will be very large (Fig 10B). Together, this means that the degree of uncertainty in the model predictions, compared to the difference in predictions between estimation and validation data, can be used to assess the novelty of the validation data. The desired scenario would be to have validation data such that the predictions are well-determined and are different between estimation and validation data (Fig 10C). A general approach for determining prediction uncertainty has been outlined in previous work [31] and has been implemented here for $^{13}$C MFA models in the EMU framework. A detailed description of this implementation is provided in Materials and methods, Section 4.4.

Another aspect that is important to consider for the case of $^{13}$C MFA, if the validation data consists of MIDs from a new tracer experiment, is that the new tracer is suitable to be used for validation. One approach to ensure that the new tracer generates data that is truly independent of the estimation data is to perform an EMU basis vector analysis [32]. This approach ensures that the tracers for the estimation and validation data produce linearly independent EMU basis vectors, which guarantees that the experiments give complementary information. This also ensures that one avoids the pitfall of having the validation data containing the same information as the estimation data, i.e. that the validation data is too similar to the estimation data.

To demonstrate that the validation data used in the simulated $^{13}$C MFA example above does not fall into these pitfalls, the prediction uncertainties for the chosen model structure $\mathcal{M}_4$ has been determined (Fig 11). As can be seen, the prediction uncertainties (light blue bars' error bars) are well determined for all MIDs. We have thus avoided pitfall 2 above, i.e. the validation data is not too novel. We have also avoided pitfall 1, since i) the predicted MIDs (light blue) are non-overlapping with the estimation data MIDs (red bars), for many of the MIDs, ii) the EMU basis vectors are linearly independent.

## 2.5 Model selection on cultured epithelial cells

Finally, we applied validation-based model selection on data from batch cultures of human cells. We performed two isotope labelling experiments with immortalized human mammary epithelium cells (HMECs), cultured with either U-$^{13}$C-glucose or U-$^{13}$C-glutamine for 6 cell doublings to achieve isotopic and metabolic steady-state (Materials and Methods, Section 4.6). MID data for nine metabolites were used as measurement data for this example, and the model structures used were the same as those presented previously in Section 2.3. The sample standard deviations from biological replicates were very small, around $s = 0.005$. In contrast to

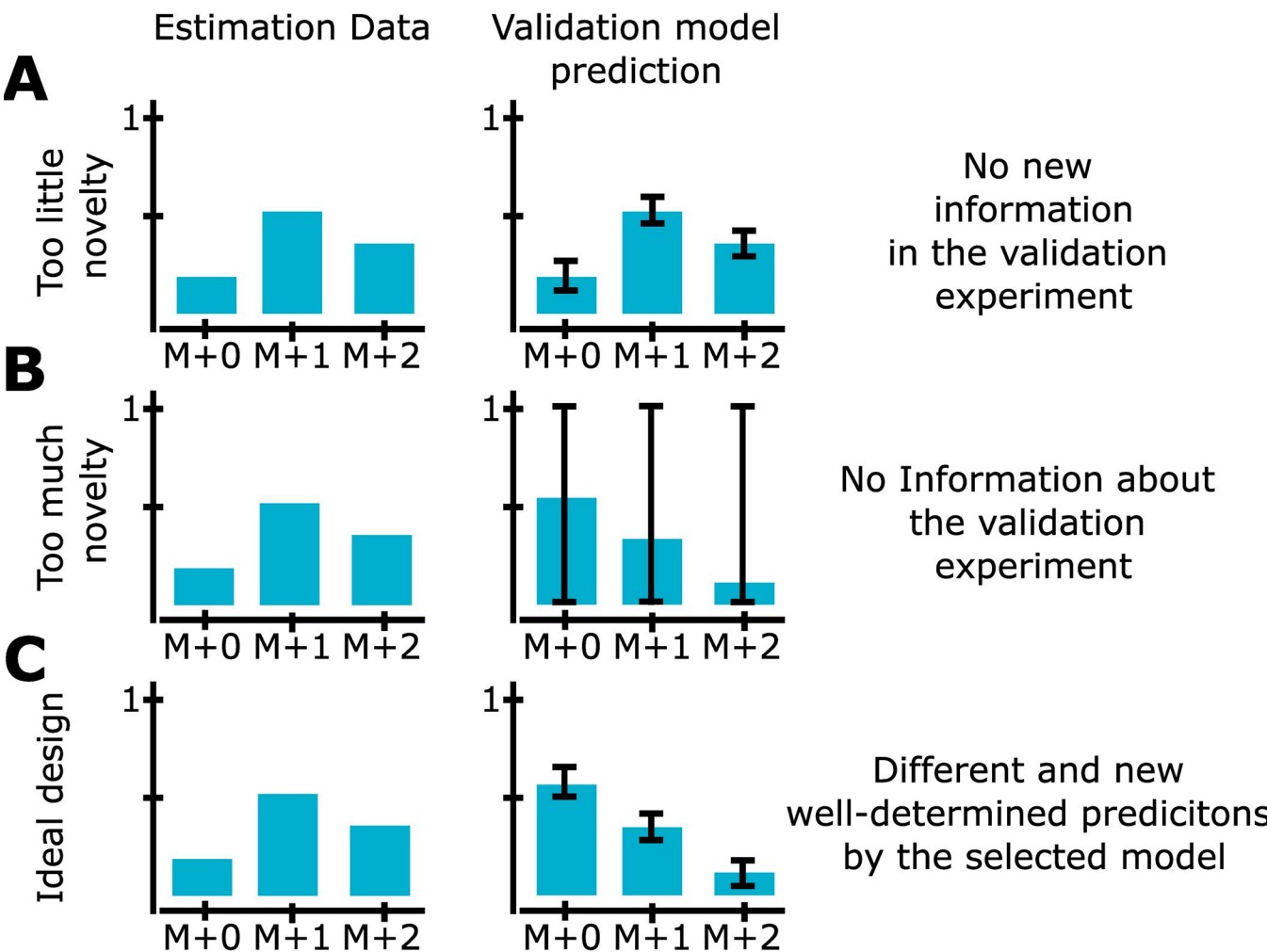

**Fig 10. How prediction uncertainty can be used to assess the novelty in the validation data.** (A) If there is too little novelty in the validation data, differences between estimation data and validation data will typically be smaller than the prediction and measurement uncertainty. (B) If there is too much novelty in the validation data, there is no information about the corresponding MIDs, and the prediction uncertainty will be large, approaching [0,1]. (C) An ideal design of validation data is thus to have well-determined predictions that are different compared to the estimation data. To be sure that there really is new information, one should also check that the new fluxes generate linearly independent EMU basis vectors (Section 2.4).

the previous theoretical examples for this system, the true $\sigma_r$ and the true model structure is now unknown. However, by evaluating the six model selection approaches for a range of different believed $\sigma_b$, it is clear that the results are consistent even for these data (Fig 12). The "SSR" method always chose the most complex model ($\mathcal{M}_7$, Fig 12A). The "First $\chi^2$" method selected model $\mathcal{M}_4$ for $\sigma_b\approx0.03$ (Fig 12B), while for the smaller $\sigma_b$, no model passed the $\chi^2$-test. The "Best $\chi^2$" method selected model $\mathcal{M}_6$ for $\sigma_b\approx0.3$, 0.015, 0.003 (Fig 12C). Similarly, the BIC approach selects model $\mathcal{M}_6$ for all values of $\sigma_b$,while the AIC selects model $\mathcal{M}_6$ for $\sigma_b\approx0.03$ and 0.015, and model $\mathcal{M}_7$ for $\sigma_b = 0.003$ (S5 Fig).

Similarly, the validation-based approach selected $\mathcal{M}_6$, regardless of $\sigma_b$ (Fig 12D). Model $\mathcal{M}_6$ excludes reactions for unlabeled acetyl group entry into acetyl-CoA (included in $\mathcal{M}_7$), which represents catabolism of pre-existing fatty acids or acetate. Hence, the choice of $\mathcal{M}_6$ suggests that such entry does not occur in these cultures. This seems reasonable since the

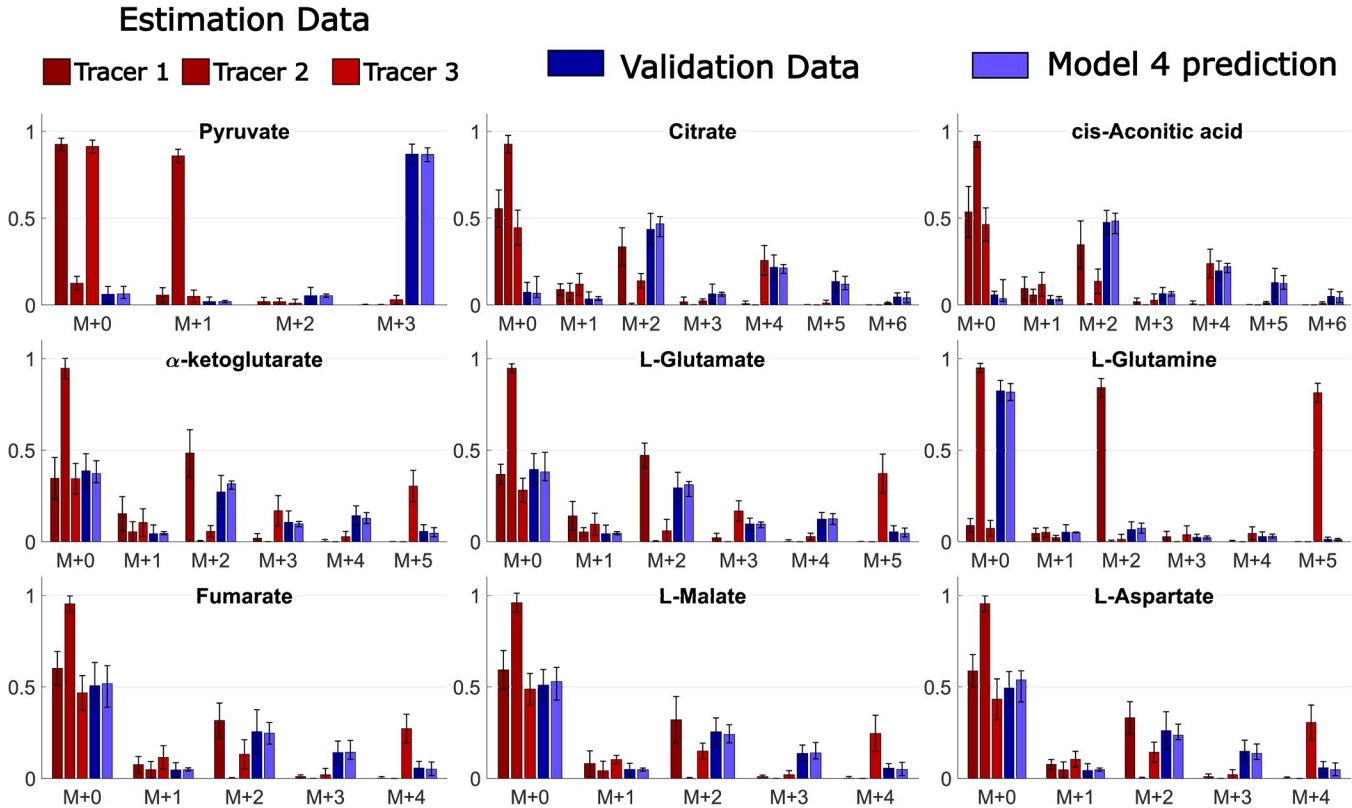

**Fig 11. Usage of prediction uncertainty to demonstrate that the validation data has neither too little, nor too much, novelty, compared to the estimation data.** This analysis shows the result from the simulated $^{13}$C MFA example (Fig 7–9). The model was trained on estimation data corresponding to three tracers: Tracer 1 = 1,2-$^{13}$C-glutamine (dark red), Tracer 2 = 3-$^{13}$C-pyruvate (red), and Tracer 3 = U-$^{13}$C-glutamine (light red). The validation data (dark blue) came from usage of tracer U-$^{13}$C-pyruvate. For the experimental data, the error bars represent standard deviation, and for the model predictions (light blue), the error bars represent model uncertainty (Section 4.4).

culture medium did not contain acetate, and was also free from serum and therefore contained very little fat. On the other hand, $\mathcal{M}_6$ includes the pyruvate carboxylase reaction while $\mathcal{M}_5$ does not, suggesting that this reaction was necessary to explain the data. Also, the pyruvate carboxylase flux was nonzero (95% confidence interval [0.08 0.98]). Interestingly, pyruvate carboxylase has been shown to be present in mammary epithelium *in vivo*, where it is important for *de novo* fatty acid synthesis [33] by replenishing the TCA cycle carbon that is consumed by citrate export ("anaplerosis"). To investigate if fatty acid synthesis also occurred in the cultured HMEC cells, we measured the MID of cellular lysophosphatidylcholine (LPC) 16:0 as a proxy for palmitate, which was not detectable with the methods used, after 7 days of $^{13}$C labeling (Fig 13). The observed MID indicated that LPC 16:0 was a mixture of $^{13}$C-labeled and unlabeled species, with higher mass isotopomers indicating that fatty acid synthesis indeed occurred. To further test the selected model structure $\mathcal{M}_6$, we used the estimated MID for cytosolic acetate (Fig 13B) from the fitted model to predict the MID of palmitate and LPC 16:0, assuming a linear mixture of pre-existing (unlabeled) and newly synthesized (labeled) species (Fig 13A). We found a reasonably good fit to the observed MID data at 82% newly synthesized LPC 16:0, indicating that the selected model $\mathcal{M}_6$ reflects actual lipid metabolism in this model system (Fig 13D).

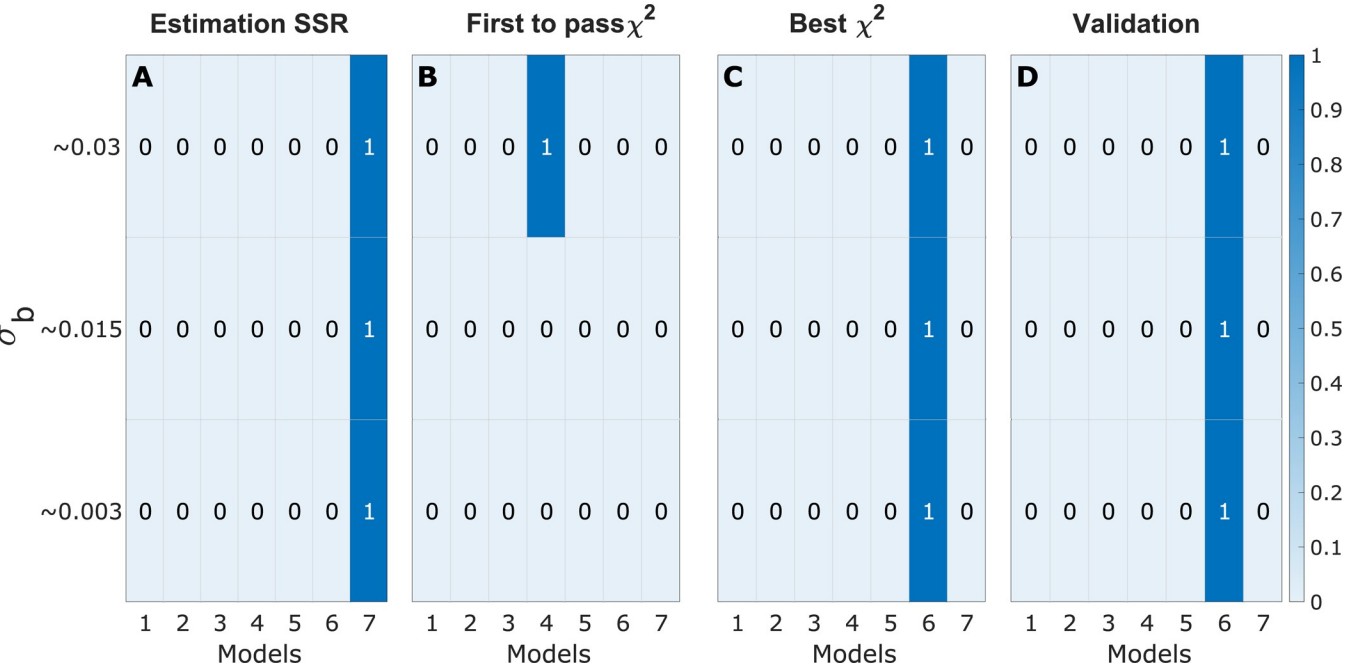

**Fig 12. Model selection results for the cultures epithelial cell example.** (A–D) Heatmaps represent results from the indicated selection methods, where rows represent different values of $\sigma_b$ and columns represent the MFA models $\mathcal{M}_1, \ldots, \mathcal{M}_7$. For each row, color indicates the fraction of times a model is selected for the given $\sigma_b$, out of 1000 samples, as indicated by the color scale (right).

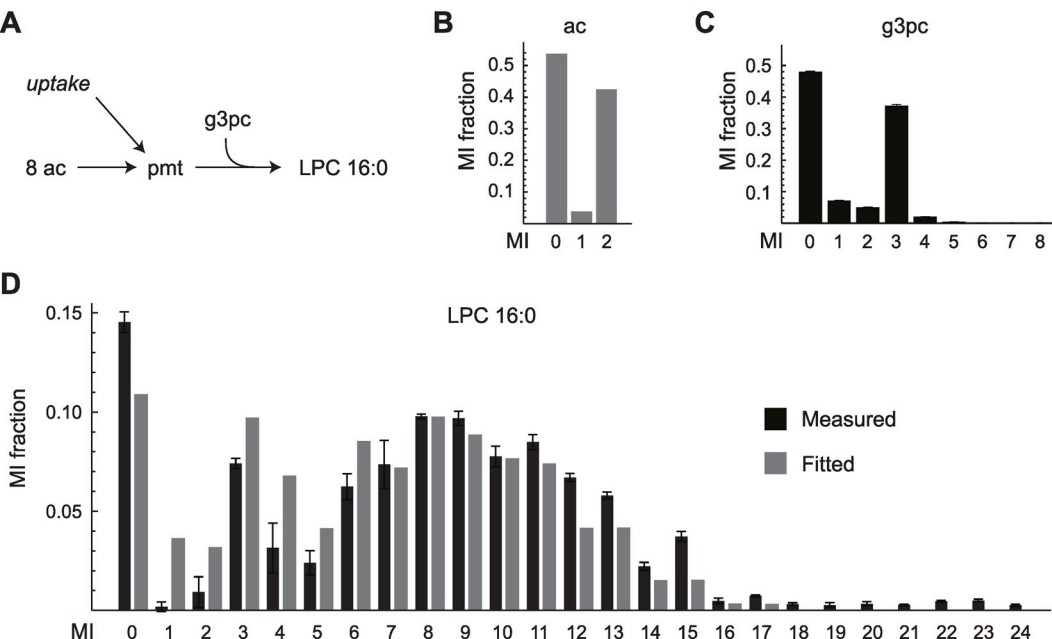

**Fig 13. Validation of lipid synthesis in HMEC cultures.** (A) Schematic of the model for lysophosphatidylcholine (LPC) 16:0 synthesis from acetate (ac). (B) Predicted MID of ac from the model selected by the "Validation" method. (C) Measured MID of glycerol-3-phosphocholine (g3pc). (D) Fitted (gray) and measured (black) MID of LPC 16:0. Mean values of biological triplicates are shown in (C, D). Error bars indicate standard deviation.

## 3. Discussion

Since estimation of metabolic fluxes using $^{13}$C MFA critically depends on the metabolic network model used, a systematic approach to model selection is of great importance. As we have demonstrated, commonly used model selection criteria such as the $\chi^2$-test can give unpredictable results if the measurement error model is not accurate. Generally, we find that standard model selection methods that rely on a compensation for model complexity will choose different models depending on the "believed" standard deviation $\sigma_b$ both for polynomial (Fig 4), linear model (Fig 6) and non-linear MFA models (Fig 8). Hence, when $\sigma_b$ is inaccurate, these methods will over- or underfit the data, which naturally leads to errors in the estimated fluxes (Fig 9). Herein, we suggest remedying this problem by performing model selection on independent validation data, which is not used for estimating model parameters ("Validation" method). From our simulation studies, it is clear that this validation-based selection method indeed is more robust and selects the correct model in a way that is independent of errors in the size of $\sigma_b$ (Figs 4, 6 and 8). Further, we demonstrate the importance of analysing the model's prediction uncertainty in order to generate confidence that the selected model accurately approximates the true metabolic system (Figs 10–11) Finally, to illustrate the potential with validation-based model development in MFA, we also applied it to new experimental data. For this data, the validation-based method consistently identifies a single model structure, whereas "traditional" methods that exclusively rely on estimation data again select different models depending on $\sigma_b$ (Fig 12). Furthermore, we also support the choice of model by predicting the MID of LPS 16:0 with reasonable accuracy (Fig 13), which is synthesized as a result of the combination of factors that differentiates the selected model structure and the other alternatives. This illustrates yet another usage of validation data for model testing and model selection. In summary, validation-based model selection offers a more reliable approach to MFA model development when measurement errors are uncertain.

There are several reasons why the model (Eq (5)) of normal-distributed, independent errors may not be accurate for MID data. First, the since mass isotopomer fractions are constrained within [0,1] and sum to 1, strictly speaking they cannot be normal-distributed, nor independent. The normal assumption is particularly inaccurate for values close to 0 or 1, where the variance becomes very small. A better noise model for MIDs might be log-normal or other distributions on the n-simplex [16]. Moreover, MI fractions obtained from mass spectrometry can be biased for technical reasons: peak integration methods can affect MID accuracy [34], and minor isotopomers may underestimated due to limited sensitivity [15]. Finally, there are biological sources of error that are difficult to avoid. For example, in batch cultures, cells can never attain perfect metabolic steady-state, and there may be unforeseen kinds of compartmentalization, such as cell subpopulations, organelles, or reaction channeling [35]. Taken together, the result of these "hidden" factors is that observed standard deviations $s$ will be artificially small compared to the residuals $y_i - \hat{y}_i$. While it could be argued that such biases constitute model error, and that the $\chi^2$-test is correct in rejecting such models, it may be unrealistic to expect a perfect model fit in every scenario. Indeed, in many cases the estimated $s$ is so small that it is exceedingly difficult to find a model that passes a $\chi^2$-test, even for minor deviations from the error model. An important topic for future research is to address these issues by developing more suitable error models for MFA. However, in the meantime, validation-based model selection could offer a pragmatic way forward.

The rational for why the new validation-based method is robust with respect to errors in the magnitude of $\sigma_b$ comes from general theory from the field System identification. This theory says that if the data has been generated by a "true" model structure $\mathcal{M}_0$ for some "true" parameters, $\theta_0$, the estimated parameters will converge to $\theta_0$ as the number of data points goes

to infinity. This theory assumes that the data used for model training is informative, i.e. that one would not gain any information regarding model distinction by exciting the system further, and that there is no redundancy of parameters, as is the case e.g. in structurally unidentifiable models. The convergence of the parameters to $\theta_0$ holds for a large class of nonlinear model structures, which include all examples considered herein [36]. Furthermore, the overall magnitude of $\sigma_b$ can be broken out from the cost function, and will thus not impact the location of any minima. These two facts together means that if the magnitude of sigma is the only thing that is wrong with sigma, the true model structure will still converge to the true parameters, while a too simple or too complex model structure will converge to the wrong parameters. This simple observation is the underlying motivation behind the validation-based method. Finally, note that the underlying theory from System identification assumes that in practice, the magnitude or scaling-error of $\sigma_b$ is not necessarily the exact same for all datapoints, and the results herein indicate that the new validation-based method is a good choice also in such situations. In other words, the validation-based method is useful also in cases when the error in the believed $\sigma$ is not homogeneously scaled for all data points. Furthermore, note that if one knows that the magnitude of the uncertainty for one metabolite is different form another metabolite, $\sigma_b$ can still reflect this difference, and the presented results will still hold as long as the same difference in magnitude between metabolite measurement uncertainties also is the case for the true measurement uncertainty, $\sigma_r$. Finally, we believe that a validation-based approach is beneficial also in situations where $\sigma$ is completely unknown, since a model that successfully predicts independent validation data probably is a decent description of reality. Note however that if the believed value of sigma, $\sigma_b$, is scaled wrong for some datapoint but not for others, the parameter estimation will be biased towards those datapoints and will converge to the wrong parameters. In this case, the predictions will be wrong, and the validation method may select the wrong model structure.

A key issue with the new validation method concerns how one divides data into an estimation and a validation data set. Clearly, the validation data must contain truly "novel" data: it is not sufficient to merely divide up replicate measurements y from the same experiment, which only differ by random noise. Herein, we have always used independent experiments with different inputs (tracers) $x$ for validation data. In this form, validation-based model selection for MFA requires parallel experiments with distinct tracers, which naturally increases the experimental effort. An alternative might be to reserve certain measurement components $y_i$ for the validation data set. In principle, the same methods for calculations of prediction uncertainty and validation-based model selection (Section 2.4) should be applicable also then, and preliminary analysis shows that this is indeed the case (S6 Fig).

The issue of which data points to reserve for the validation set is more difficult in our setting than in traditional cross-validation over statistically independent samples from a fixed data distribution. On one hand, highly dissimilar data points will be more difficult to predict, and should therefore provide a more stringent test for model selection. On the other hand, too dissimilar validation data ("extrapolation") may not be predictable by any model. To judge this tradeoff, the prediction uncertainty method in Section 2.4 is useful. This topic is also relevant for experimental design, which could be adapted to generate data suitable for informative validation data. Finally, an interesting aspect of these results is that also too simple models have a too large prediction uncertainty (S7 Fig), which is contrary to the traditional principle of bias-variance tradeoff, which says that only too complex models have a too high variance. This further emphasizes the fundamental differences between statistical methods based on sampling from the same distribution, compared to methods for mechanistic modelling, where data from different distributions can be used for the validation analysis. These results argue for a revision of such previously established truths, coming from statistics [13,31,37].

It is important to distinguish between model selection and model testing. As mentioned earlier, while our method allows selecting the best model from a given set, it does not guarantee that the best model is indeed acceptable. While a goodness-of-fit test could be performed on the validation data for the selected best model, such a test will in general be optimistic due to multiple testing over models. For proper model testing, a third "test" data set should be used, which is not used for either parameter estimation of model selection.

The analysis in Fig 8 is only meant to illustrate that errors in model structure can lead to errors in flux estimation, and is not an exhaustive analysis of what these errors might look like. The fluxes depicted in Fig 8 is considered a representative selection of the fluxes that overlaps between all model structures. The overall fact that errors in model structure may lead to more or less large errors in flux estimations should hold true.

Based on our results, we suggest that validation-based model selection should always be considered when developing MFA models. Nevertheless, for small errors in $\sigma_b$, the less computationally expensive methods, such as AIC and BIC may give the same results. The problem, however, is that one does not know when the dependency on errors in $\sigma_b$ make those methods unreliable in cases were the magnitude of the experimental error is uncertain, and in such cases it is therefore safer to use a validation-based approach. Validation-based approaches also have important advantages related to interpretation, and are therefore common-place in other field.

We believe that the field of MFA modelling should take inspiration from such other fields of computational biology, where the ability to correctly predict independent data *not* used for parameter estimation is a standard criterion for model quality, and where such validation tests often are a requirement for publication [20,21,23,24,26,27]. Given that models are always simplifications of reality, such independent validation is important both for the modelling process and for communicating results to non-experts users. In other words, while it is almost always wrong to assume steady-state metabolism occurring in a single average cell, such a model may still be a good enough approximation of reality to produce realistic fluxes. Importantly, one way to demonstrate the realism and general predictive power of the chosen model is to show that it can predict new independent validation data. Notably, in guidelines issued by the US Food and Drug Administration (FDA), testing on independent validation data is necessary condition for a model to be considered trustworthy [38]. All in all, we believe that validation-based model selection provides sound and reliable checking of metabolic models, which we hope will be of value also to the $^{13}$C MFA field.

# 4. Materials and methods

## 4.1 $^{13}$C Metabolic flux analysis

As stated previously, the gold standard method for measuring metabolic fluxes in a given system is model-based metabolic flux analysis with isotopically labelled tracers. The model $\mathcal{M}$ includes the stoichiometry and atom mappings for each reaction, and is parameterized by the metabolic fluxes $v$, or more precisely, by the independent fluxes $u$. At steady state, the model-predicted MIDs $\hat{y}$ are uniquely determined by $u$ together with the known isotope distributions $x$ of the network substrates [9,39],

$$\hat{y} = h(x, u) \tag{3}$$

where the function $h$ is determined by the model $\mathcal{M}$. Model fitting is done by seeking the vector $u$ that minimizes the sum of the squared weighted residuals (SSR) between the model-

predicted and measured MIDs [40],

$$f(u) = SSR = \sum_{i=1}^{m} \sum_{j=1}^{n} \left( \frac{\hat{y}_j^i - y_j^i}{\sigma_{ij}} \right)^2 \tag{4}$$

$$y_j^i = h_j(x^i, u_0) + \epsilon_{ij} \ \epsilon_{ij} \ \epsilon \ N(0, \sigma_{ij}) \tag{5}$$

Here each measurement $y_j^i$ is assumed to derive from the model prediction $\hat{y}_j^i$ at the true flux vector $u_0$, plus a normal-distributed noise $\epsilon_{ij}$ with standard deviation $\sigma_{ij}$. If there are several experiments with different tracers $x^i$, the sum is taken over all resulting measurement vectors $y^i$ [41] Under these assumptions, $f(u)$ follows a $\chi^2$-distribution, and so the $\chi^2$-test can be used to assess model fit [42]. If this test does not reject, the model $\mathcal{M}$ and the inferred fluxes $u$ are considered valid.

## 4.1 Construction of mathematical models: Predictors and the EMU framework

The mathematical models presented herein are formulated such that a mathematical structure describes one or more predictors $\hat{y}_i(\theta)$, given a set of model parameters $\theta$. These mathematical structures or models can exist in different forms, such as polynomial models or ordinary differential equation (ODE) models. For MFA, a common approach for model formulation is the Elementary metabolite units (EMUs) framework [34]. In short, the EMU framework allows for a decomposition of the model such that only the information necessary to calculate a desired set of MIDs remains. The metabolic network is broken down into EMU subnetworks that are used to formulate equations, of the form in Eq (6) below [43].

$$A_{n,k}(v) * X_{n,k} = B_{n,k}(v) * Y_{n,k}(x, X_{n-1}, \ldots, X_1) \tag{6}$$

where index $n$ indicates the size of the EMU network and index $k$ is used to index several networks of the same size; where matrices $A$ and $B$ contains the model structure for the fluxes $v$, which can be parameterized according using a smaller set of independent fluxes $u$; where matrices $X_{n,k}$ and $Y_{n,k}$ contains the unknown and known EMU variables, respectively; and where $x$ are the EMU variables that correspond to the system tracer [43]. In other words, for EMU models $\theta = u$.

## 4.2 Optimization of model parameters to fit the data

The objective of the parameter estimation step in any modelling problem is to minimize an objective function $f(\theta)$ which determines the agreement with data for a given the set of parameters $\theta$. The general optimisation problem, which determines the optimal parameters $\theta^*$, is formulated as

$$\theta^* = \arg\min_\theta f(\theta)$$

$$s.t. g_j(\theta) \geq 0 \ \forall j \tag{7}$$

where $g_j(\theta)$ are functions describing constraints applied to the optimization problem. Again, for $^{13}$C MFA modelling, the parameters $\theta$ are the independent fluxes, herein denoted $u$. Also,

for $^{13}$C MFA modelling, two constraints are usually placed on these independent fluxes:

$$g_1(u) : \; null(S) * u \geq 0$$

$$g_2(u) : \; ub \geq u \geq lb$$

where $null(s)$ is a null space matrix of the network's stoichiometry matrix; and where $ub$ and $lb$ are the upper and lower bounds of $u$, respectively. The first condition ensures that all fluxes, which are given by the product of $null(S)$ and $u$, are positive. The second condition ensures that the independent fluxes are constrained within a predetermined interval. As for the detailed form of the objective function, $f$, it can vary depending on the specific analysis conducted, but will generally be some variant of the weighted SSR function, since this objective function has sound theoretical properties [36]. The SSR used herein is given by Eq (4). For the EMU model, the relationships between $y$, $\hat{y}$, $\sigma$, and the state variables $X$ are given by:

$$\hat{y}_i(u) = X_{n,k}(l, m) \tag{8}$$

where $y_i$ is the measured value for the $m^{th}$ mass fraction of the $l^{th}$ EMU in $X_{n,k}$, i.e. a specific bar in Fig 1C. If $\hat{y}_i$ is a mean value of multiple original data points, then the residuals $(y_i - \hat{y}_i(u))$, should be weighted with the standard error of the mean ($SEM_i$) rather than $\sigma_i$ should be used in Eq (4). The relation between the two is given by:

$$SEM_i = \frac{\sigma_i}{\sqrt{N}} \tag{9}$$

where $N$ is the number of sample points. However, all of these are theoretical truths; which denominator to use in Eq (4) is an unresolved issue for $^{13}$C MFA models (see Section 3.2).

## 4.3 $\chi^2$-test

In the $^{13}$C MFA modelling field, the $\chi^2$-test is the statistical hypothesis test that most commonly is employed to evaluate whether the *SSR* is small enough, i.e. if the model can be considered an accurate representation of the target system. In practice, the *SSR* is compared with the inverse cumulative $\chi^2$-distribution, where the degrees of freedom is given by the number of datapoints adjusted for the fact that some independence between the datapoints and the model is lost by estimating parameters to the same data that is used for testing. This compensation can be done in different ways; the naive way is to do no compensation at all, and the most conservative way is to compensate for all parameters (free fluxes) in the model. The most accurate version is to instead use the number of practically identifiable parameters. In reality, this adjustment is done in different ways, often without justification, and these differences may be the reason why a model is, or is not, rejected. This ambiguity is one of the reasons arguing against the usage of this test. Its dependency on the value of $\sigma_i$ is another such argument. With the most conservative choice, the algorithm becomes:

Input: model structure $\mathcal{M}$, parameters $\theta$, data $D$ with $N$ datapoints.

1. Calculate the combined *SSR* for all data points $N$, using (Eq (4)).

2. If $SSR < \chi^{2,cuminv}(p = 0.95, N - \theta)$, i.e. the cumulative inverse of a $\chi^2$ distribution, then model structure $\mathcal{M}$ is accepted

   *else* model structure $\mathcal{M}$ is rejected.
   Output: FAIL if model structure $\mathcal{M}$ is rejected OR PASS if model structure $\mathcal{M}$ acceptable with respect to $D$.

## 4.4 Determining model prediction uncertainty

In this work two main approaches were used to determine the prediction uncertainties. The first and primary approach was a prediction profile likelihood (PPL) analysis. A prediction profile likelihood analysis is used to determine the uncertainty of a predicted model value or property [44–46]. The PPL-analysis was implemented by modifying the function for the SSR, seen in Eq (4), such that it contained an additional term such as:

$$f(\theta) = \sum_{i=1}^{m}\sum_{j=1}^{n} \left( \frac{\hat{y}_j^i - y_j^i}{\sigma_{i,j}} \right)^2 + W * (p_{target}^{\omega} - p_{sim}^{\omega}(\theta))^2 \tag{10}$$

where $p_{sim}^{\omega}$ is the simulated relative abundance of mass fraction $\omega$ and is determined by the parameters $\theta$. $p_{target}^{\omega}$ is a set target value for mass fraction $\omega$ and $W$ is an integer with an arbitrary large value. By assigning a very large value to $W$, any difference between $p_{sim}^{\omega}$ and $p_{target}^{\omega}$ will be magnified. Thus, the optimization process will select parameters that minimizes this difference. Then, $p_{target}^{\omega}$ was gradually stepped away from the optimal simulated value of the mass fraction $\omega$, until a cutoff value is reached. This stepping process is repeated for all mass fractions that are included in the prediction.

The second approach used for determining the model prediction uncertainty was estimation through Markov chain Monte Carlo (MCMC) sampling. For this analysis a posterior distribution of parameter values is generated, and all parameter sets that are acceptable with respect to the estimation data are collected. The model prediction uncertainty is then determined by the interval:

$$CI_{\alpha,Dof} = [f(\theta) \leq f(\theta^*) \pm \Delta_{\alpha}(\chi_{DoF}^2)] \tag{11}$$

where $f(\theta)$ is the generalise form of the SSR objective function described in Eq (4); $f(\theta^*)$ is the SSR function value for the optimal parameters; $\alpha$ is the confidence level; $\Delta_{\alpha}(\chi^2)$ is the quantile of the $\chi^2$-statistic; $DoF$ is the degrees of freedom; and $\theta^*$ are the optimal parameters. In this work, the $DoF$ is equal to the number of model parameters, i.e. in the $^{13}$C MFA examples the number of free fluxes, and $10^5$ samples were used for the sampling.

## 4.5 Simulated data generation

For the examples presented in this paper, simulated data is utilized to create scenarios for model estimation in which the ground truth is known. In each of these examples, a given model $\mathcal{M}_0$ with parameters $\theta_0$ has been used to generate values for selected variables of interest, given a predetermined set of model parameters and inputs. To these values, a normally distributed noise was added, with a given true sigma, $\sigma_r$.

For the polynomial exampled a seventh order polynomial were used as true model (see S2 Table, for true parameter values), and a normally distributed noise was added with $\sigma_r = 0.2$. The model was then fitted to data corresponding to one realisation of this noise i.e. $i = 1$, using Eq (4).

For the linear model example, the model $A_3$ (Fig 5) was used as the true model (see S2 Table for true parameter values). A normally distributed noise was added with a true sigma of $\sigma_r = 5$ The model was then fitted to data corresponding to five realisations of this noise i.e. $i = 5$, using Eq (4).

For the EMU-model example, model 4 (Fig 7) was used as the true models (see S2 Table for true parameter values). A normally distributed noise was added with a true sigma of $\sigma_r = 0.03$ The model was then fitted to data corresponding to 3 realisations of this noise i.e. $i = 3$, using

Eq (4). The data was generated from four different tracers. These tracers were U-$^{13}$C-pyruvate, U-$^{13}$C-glutamine, 3-$^{13}$C-pyruvate, and 1,2-$^{13}$C-glutamine. For the "Validation" method the data from tracers U-$^{13}$C-glutamine, 3-$^{13}$C-pyruvate, and 1,2-$^{13}$C-glutamine was used as $D^{est}$ while data from U-$^{13}$C-pyruvate was used as $D^{val}$

To generate the different values of $\sigma_b$, the sample standard deviation for each observation was scaled by a number drawn from a uniform random distribution. For $\sigma_b \approx 10^* \sigma_r$ the distribution range was between [8 12], for $\sigma_b \approx \sigma_r$ the distribution range was between [0.8 1.2], and for $\sigma_b \approx 0.1^* \sigma_r$ the distribution range was between [0.08 0.012].

## 4.6 Cell culture and isotope tracing

Human Mammary Epithelial Cells (HMECs) were obtained from the laboratory of William C. Hahn (Dana-Farber Cancer Institute, Boston, USA) and have been previously described [47]. HMECs were grown in custom-synthesized Mammary Epithelial Basal Medium (MCDB) 170 [48] supplemented with 1% Mammary Epithelial Growth Supplement (MEGS) (S0155, Gibco), 100 units/ml penicillin and 100 μg/ml streptomycin (15140122, Thermo Fisher Scientific). Cells were kept in a humidified atmosphere of 5% $CO_2$/95% air at 37˚C and washed and detached using ReagentPack Subculture Reagents (CC-5034, Lonza).

For isotope tracing experiments, 400,000 cells were seeded at day 0 in a T25 flask in 5mL medium and incubated overnight. On day 1, medium was changed to an MCDB 170 medium of the same molar composition, but with glucose or glutamine exchanged for U-$^{13}$C-glucose or U-$^{13}$C-glutamine (Cambridge Isotope Laboratories), respectively. On day 2, each T25 flask culture was detached and seeded into in two T25 flasks, using the same medium. On day 4, cells were detached and seeded into 6-well plates at 250,000 cells/well in 2mL of medium, in triplicate for each tracer. On day 7 (after roughly 6 cell divisions in the presence of each $^{13}$C tracer), each multi-well plate was placed on ice, medium was aspirated and cells were washed twice with 1 mL of cold PBS. Then, 1 mL cold (–80˚C) methanol (JT Baker, BAKR8402.2500, VWR) was added to each well, cells were scraped using a 17mm cell scraper (83.1830, Sarstedt), and the extracts were carefully transferred to a new tube, vortexed for 30 seconds to break up aggregated cell material, and stored in -80˚C until analysis. LCMS analysis of cell extracts was performed using a pHILIC LC column coupled to a Thermo QExactive orbitrap mass spectrometer, as previously described [49]. All metabolite peaks reported were confirmed against pure standards. Peak areas were integrated directly from instrument data using the mzAccess data access framework [50] and Mathematica v.11.1 (Wolfram Research). Mass isotopomer distributions were calculated as the areas of each mass isotopomer peak, divided by the total peak area for all mass isotopomers.

## 4.7 Model for the LPC 16:0 MID

The MID $x_{ac}$ for acetate (ac) was obtained from the fitted model as described above in section 2.5. The MID $x_{pmt}$ of total cellular pool was modeled as a linear mixture

$$x_{pmt} = \alpha \, x_{pmt}^{new} + (1 - \alpha)x_{pmt}^{old} \tag{12}$$

where $x_{pmt}^{new}$ is the MID of newly synthesized palmitate (pmt), computed by convolution of $x_{ac}$ eight times, modeling the condensation of eight ac molecules by fatty acid synthase; $x_{pmt}^{old}$ is the natural MID, representing pre-existing palmitate; and $\alpha$ is the unknown mixture coefficient. The lysophosphatidylcholine MID $x_{lpc}$ was modeled as a convolution of $x_{pmt}$ and the measured glycerol-3-phosphocholine (g3pc) MID $x_{g3pc}$. This can be written as:

$$x_{lpc} = C(x_{g3pc})x_{pmt} \tag{13}$$

where $C(x_{g3pc})$ is a matrix whose elements depend on $x_{g3pc}$. This yields an equation system linear in the unknown $\alpha$, which was solved using the least-squared method.

### 4.8 Software

All analysis presented here were performed in MATLAB by The MathWorks Inc., release 2020b. For the models presented in this paper the open source MATLAB toolbox OpenFLUX2 [51] was employed to transform the network structure to the EMU equation systems.

## Supporting information

**S1 Fig. Model selection results for the polynomial model example.** Heatmaps represent results from the AIC (left) and BIC (right) methods, where rows represent different values of $\sigma_b$ and columns represent the polynomial models $h_1,\ldots,h_{14}$. For each row, colour indicates the fraction of times a model is selected for the given $\sigma_b$, out of 10,000 samples, as indicated by the colour scale (right).
(EPS)

**S2 Fig. Model selection results for the linear model example.** Heatmaps represent results from the AIC (left) and BIC (right) methods, where rows represent different values of $\sigma_b$ and columns represent the polynomial models $A_1,\ldots,A_6$. For each row, colour indicates the fraction of times a model is selected for the given $\sigma_b$, out of 1 000 samples, as indicated by the colour scale (right).
(EPS)

**S3 Fig. Illustration of the data and simulation for the Linear example.** The simulated data for the linear example plotted with the different input vectors x along the x-axis and the model output y on the y-axis. The three different output variables are indicated by the different colours, $y_1$–purple, $y_2$–Green, and $y_3$–orange. The division into estimation data (left) and validation data (right) is indicated by the vertical line.
(EPS)

**S4 Fig. Model selection results for the simulated 13C MFA model example.** Heatmaps represent results from the AIC (left) and BIC (right) methods, where rows represent different values of $\sigma_b$ and columns represent the polynomial models $\mathcal{M}_1,\ldots,\mathcal{M}_7$. For each row, colour indicates the fraction of times a model is selected for the given $\sigma_b$, out of 100 samples, as indicated by the colour scale (right).
(EPS)

**S5 Fig. Model selection results for the epithelial cell example.** Heatmaps represent results from the AIC (left) and BIC (right) methods, where rows represent different values of $\sigma_b$ and columns represent the polynomial models $\mathcal{M}_1,\ldots,\mathcal{M}_7$.
(EPS)

**S6 Fig. Usage of Validation data with prediction uncertainty where a sub part of a single data set has been reserved for validation.** This preliminary analysis shows the result of using validation with prediction uncertainty in a scenario where a portion of a complete data has been reversed as validation data. In this example the model was trained on estimation data (dark red) consisting of MIDs from 8 metabolites, from two tracers, with the model fit to the estimation data (light red) is illustrated for each MID. The validation data consisted of the MID for α-ketoglutarate (dark blue) and the model prediction is illustrated (light blue) and shows good agreement with the validation data. For the experimental data, the error bars represent standard deviation, and for the model predictions, the error bars represent model

uncertainty (Section 4.4). The tracers are Tracer 1 = U-$^{13}$C-glutamine, Tracer 2 = U-$^{13}$C-pyruvate.
(EPS)

**S7 Fig. A comparison of model predictions with uncertainties compared to the validation data for the simulated $^{13}$C MFA example.** The validation data for the simulated $^{13}$CMFA example consisted of MIDs for nine metabolites from a [U-13C] pyruvate tracer. The metabolites are from top-left to bottom-right, pyruvate, citrate, cis-aconitic acid, alpha-ketoglutarate, L-glutamate, L-glutamine, fumarate, L-malate, and L-aspartate. The purple bars indicate the simulated MIDs, and the corresponding error bars indicate the experimental uncertainty of $\sigma_r$. The red bars indicate the predicted MIDs of model structure $\mathcal{M}_4$ and the corresponding error bars indicate the prediction uncertainty. The green bars indicate the predicted MIDs of model structure $\mathcal{M}_2$ and the corresponding error bars indicate the prediction uncertainty.
(EPS)

**S1 Algorithm. Algorithm used for the model selection problem.** The model selection algorithm takes a set of model structures and a set of data as inputs and selects the most appropriate model structure based on the sub type (A-D) and the data. Subtype A selects the model structure that yields the smallest summed squared residuals (SSR) given the entire data set. Subtype B selects the first/simplest model structure that can pass a $\chi^2$-test. Subtype C selects the model structure that passes a $\chi^2$-test with the largest margin. Finally, subtype D selects the model structure that yields the lowest SSR with respect to a validation subset of the data.
(DOCX)

**S1 Table. Breakdown of reactions for the TCA-cycle models.** The table contains a detailed breakdown of the reactions that are included in the TCA-models. In general, the model structures are derived from the most complex model structure $\mathcal{M}_7$ by successively removing and combining reactions and thus make each successive model structure simpler.
(DOCX)

**S2 Table. A summary of the parameter values that were used to generate the simulated data for the three different examples that are used in the manuscript.** The full parameter vector for the polynomial, linear and metabolic flux analysis model examples are given by the respective columns.
(DOCX)

## Author Contributions

**Conceptualization:** Nicolas Sundqvist, Roland Nilsson, Gunnar Cedersund.

**Data curation:** Nicolas Sundqvist, Nina Grankvist, Jeramie Watrous, Jain Mohit.

**Formal analysis:** Nicolas Sundqvist, Jeramie Watrous, Jain Mohit, Roland Nilsson.

**Funding acquisition:** Roland Nilsson, Gunnar Cedersund.

**Investigation:** Nicolas Sundqvist, Roland Nilsson.

**Methodology:** Nicolas Sundqvist, Roland Nilsson, Gunnar Cedersund.

**Project administration:** Gunnar Cedersund.

**Resources:** Nina Grankvist.

**Software:** Nicolas Sundqvist.

**Supervision:** Roland Nilsson, Gunnar Cedersund.

**Validation:** Nicolas Sundqvist, Roland Nilsson, Gunnar Cedersund.

**Visualization:** Nicolas Sundqvist.

**Writing – original draft:** Nicolas Sundqvist, Roland Nilsson, Gunnar Cedersund.

**Writing – review & editing:** Nicolas Sundqvist, Roland Nilsson, Gunnar Cedersund.

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
