## [Decision Letter · Decision Letter 0]

6 Sep 2021

Dear Mr Sundqvist,

Thank you very much for submitting your manuscript "Validation-based model selection for 13C metabolic flux analysis

with uncertain measurement errors" for consideration at PLOS Computational Biology.

As with all papers reviewed by the journal, your manuscript was reviewed by members of the editorial board and by several independent reviewers. In light of the reviews (below this email), we would like to invite the resubmission of a significantly-revised version that takes into account the reviewers' comments.

We cannot make any decision about publication until we have seen the revised manuscript and your response to the reviewers' comments. Your revised manuscript is also likely to be sent to reviewers for further evaluation.

Sincerely,

Vassily Hatzimanikatis

Associate Editor

PLOS Computational Biology

Kiran Patil

Deputy Editor

PLOS Computational Biology

Reviewer's Responses to Questions

**Comments to the Authors:**

Reviewer #1: The authors propose an approach to select which metabolic model to use for 13C-metabolic flux analysis (13C-MFA). The basic idea is to perform multiple tracer experiments with different isotopic tracers and then set one of these data sets aside to be used as validation data. The remaining data sets are then used to estimate fluxes using different metabolic models with increasing complexity. The hope is that the validation data will allow proper model selection, that is, selection of a metabolic model that is not too simple or too complex.

The authors are correct that model selection can have a significant impact on the fluxes that 13C-MFA produces, however, I disagree with the proposed approach for several fundamental reasons.

Major concerns:

1) The authors completely ignore the most important step in 13C-MFA, which is the design of tracer experiments. In this step, the key objective is to select the most informative isotopic tracers and isotopic labeling measurements for a given metabolic network model. In this step, the metabolic model must represent the best current knowledge of the biochemistry of the cell that is studied, i.e. the model should be as comprehensive as possible to avoid any biases in the 13C-MFA studies. The approach proposed by the authors ignores the importance of proper tracer selection. If poor tracers are selected, then it doesn’t matter what method is used for model selection, the flux results will be poor. The authors must evaluate what impact good tracer selection and poor tracer selection has on their proposed approach.

2) The proposed approach only really works if the validation data contains truly “novel” data that is independent of the other data. In reality, this can never be satisfied. How do you determine to what extent the validation data is independent? This is an important question that is never quantitatively addressed by the authors. The fact that a different tracer is used doesn’t automatically mean that the validation data is independent.

3) On the other extreme, the validation data must not be too dissimilar from the other data sets, or the validation data may not be predictable by any model. While this problem is briefly mentioned by the authors, is not fully addressed. This is a serious problem of the proposed approach. How do you ensure that the validation data is not too similar but also not too dissimilar compared to the other data? Unless the authors can provide some quantitative metrics that can be followed to quantify the degree of similarity and dissimilarity of the validation data, I am afraid that I cannot recommend the use of the proposed approach.

Reviewer #2: The paper addresses the topic of model selection in metabolic flux analysis with isotope labeling data gathered under metabolic steady state conditions (termed MFA) when the labeling data error is largely unknown, but consistent across the data set. The authors compare different existing model selection approaches (SSR, first Chi2, best Chi2, AIC, BIC, validation) by means of simulated examples (polynomials, linear toy and nonlinear MFA systems) and apply them for the evaluation of a real data set in human epithelial cells. In the latter case, they found evidence for the pyruvate carboxylase as a necessary model component. The authors argue that model selection based on validation is a more reliable approach than SSR- and Chi2-based metrics in case of unknown data standard deviations. A conclusion about AIC/BIC is not drawn.

Overall, the work addresses a relevant, up to now underrepresented topic in the field of MFA. Some major issues concerning the precise aims/scope/results of the work need to be addressed. If done, the manuscript has the potential to become fit for publication in PLOS Comput Biol after some major revisions.

**Major Comments**

1. Model selection is always connected to a goal. The authors show examples for flux estimation, pathways inference and label prediction. Is the intention of model selection in this work directed towards all of these three categories? That this is not clearly stated is a weak spot of the manuscript; and it makes it somewhat hard to assess whether there is substantial evidence for the conclusions.

2. The presented results achieved across the examples, indicate that the performance of comparably cheap criteria (AIC, BIC) appear to be on par with the validation-based (VB) approach for all cases for which they are reported (Why do the authors calculate model selection AIC/BIC criteria for all, but the epithelial cell example?). Interestingly, this is not discussed. In addition, VB is computationally expensive. An overall comparison of criteria performances should be added incl. a discussion of the cost/benefit aspect.

3. Following up Comment #2, the AIC is an approximation of leave-one-out cross-validation according to Gelman's Bayesian Data Analysis. Could then the AIC in theory not even be better than the proposed VB scheme?

4. The Best Chi2 approach performs in some cases better than VB (in terms of fractions), in particular for the case of approx. correct and larger errors. Can the authors comment on this behavior? Together with Comments #1,2 above, this may find its way into a more fine-grained advice to modelers that apply model selection in cases of known vs. grossly known vs. unknown model errors rather than suggesting the consideration of VB per se (P19).

5. Please add some references on the origins of the epithelial network model and on which basis the model variants are constructed. A network description directly attached to this manuscript would aid understanding and bridge between the thumbnail-sized pictures and the reaction names discussed in the text (see also below).

**Minor Comments**

1. Abstract: "These errors are often not known in practical examples." This statement suggests that it is standard that nothing is known about data errors in the field of MFA. This is certainly a disputable claim that may snub analytical chemists working in this field. Arguably, there is lots of work where approx. errors are formulated on well-grounded experience. Therefore, this reviewer suggests that the authors revise their statement appropriately.

2. P4 "s does not account for experimental bias, such as deviations from metabolic steady-state that always occur in batch cultures." This statement is difficult because if the MSS considerably questioned, applying MFA is doubtful. Moreover, wouldn't repetition of the experiment help to (un)cover such deviations at metabolic level?

3. The authors speculate about the approaches that are applied in the MFA domain for model selection. Indeed the procedure described in (Antoniewicz 2018) may be interpreted as First Chi2 approach. The iterative scheme sketched in (Dalman 2016) points more to the use of the SSR/Chi2 as a means for model validation/testing and in this sense would have been correctly applied. Generally, it would improve clarity if the authors introduce the differences between model validation/testing and model selection in the introduction, rather than mentioning it on the last page, and take care distinguishing these two traits in commenting the literature.

4. P6: "For all examples herein, data from distinct model inputs is used for validation." Does this mean data from tracer experiments with different tracer species? Please clarify.

5. Sec. 2.1: The motivating example, although being a simple illustration to unfamiliar readers, is clearly outside the main story line. Thus, this reviewer suggests to consider shifting the example to the supplement, and keep focused on labeling systems.

Also, here the difference between model validation/testing and selection becomes evidently mixed (see also #3): the most simple and most complex polynomials will certainly fail in a model validation/testing step.

4. P11: Please report the number of resamplings in the main text, rather than just in the caption of Fig. 8.

6. P13/Fig. 9: Please report the full name of the reactions discussed to "link" the results better to the metabolic network context (see also Major Comment #5).

7. Sec. 2.4 Please report also AIC/BIC numbers (see also Major Comment #2).

8. P15: "suggesting that fatty acid synthesis indeed occurred". Please clarify whether there is potentially another route than FA synthesis for label entering this compound (see also Major Comment #5).

9. Fig. 11: Please add error bars to the mean MIDs.

10. P17: "correctly predicting" brings up the question on how correct, correct is. The more qualitative wording used on P15 "reasonable good fit" appears to be better suited here.

11. P18: "In practice, the magnitude or scaling-error of is not necessarily the same for all data points, and the results herein indicate that the new validation-based is a good choice also in such situations". To the understanding of the reviewer, this has not been tested. Please comment.

12. For clarity, please use consistent names for the different test cases throughout text and figures, e.g. "simulated 13C-MFA model example" in Fig 7+8+9, or "cultured epithelial cells" in Fig. 10

**Typos/Wording**

P2 This problem is central

P3 Correct system hypothesis  System hypothesis

P4 approached  approach; passes  pass; are often below  are often reported to be below (add reference?); former alternative leads to high uncertainty  former alternative may lead to high uncertainty

P9 simulate data form ( from) 6 distinct input vectors

P14 tested  applied

P23 check notation: M0 model with theta0 parameters is selected for data generation, with M0=M7 in case of the polynomial example, M0=A3 in case of ... etc.

**Have the authors made all data and (if applicable) computational code underlying the findings in their manuscript fully available?**

Reviewer #1: Yes

Reviewer #2: Yes

PLOS authors have the option to publish the peer review history of their article (what does this mean?). If published, this will include your full peer review and any attached files.

Reviewer #1: No

Reviewer #2: No
---

## [Decision Letter · Decision Letter 1]

5 Jan 2022

Dear Mr Sundqvist,

Thank you very much for submitting your manuscript "Validation-based model selection for 13C metabolic flux analysis

with uncertain measurement errors" for consideration at PLOS Computational Biology. As with all papers reviewed by the journal, your manuscript was reviewed by members of the editorial board and by several independent reviewers. The reviewers appreciated the attention to an important topic. Based on the reviews, we are likely to accept this manuscript for publication, providing that you modify the manuscript according to the review recommendations.

Sincerely,

Vassily Hatzimanikatis

Associate Editor

PLOS Computational Biology

Kiran Patil

Deputy Editor

PLOS Computational Biology

[LINK]

Reviewer's Responses to Questions

**Comments to the Authors:**

Reviewer #2: The minor comments below are mainly related to sharpen statements that otherwise risk being misunderstood by the target community. Since line numbers are missing, page numbers are given.

Minor Comments

1. Reply to Response to Major Comment 5:

While the list given in the SI about model variants is clearly useful as an overview, this way is clearly insufficient for reproducing results or even a simple model simulation step. Please take care to adhere to the reporting standards established in the field.

2. Reply to Response to Major Comment 4 and Minor Comment 1:

Abstract, Summary, P15: This reviewer agrees that estimating the precise true error is hardly possible. Indeed, results presented indicate that gross mis-characterization of these errors leads to non-robust model selections, which in turn CAN lead to errors in flux estimates. But, in standard use-cases, such as E. coli and GC-MS, the order of the errors in the labeling data is indeed well characterized. Here, the results presented do indicate that if the true error is approximately correctly known, traditional methods work fairly well *and* are computationally efficient. Thus, validation-based model selection is useful in cases for which the magnitude of the errors is unknown or a gross mis-characterization is suspected. Please be precise here and adequate in the conclusions.

3. Reply to Response to Minor Comment 2:

Certainly, modeling assumptions are always idealizing the truth. The fundamental principles and experimental requirements for 13C MFA are well-known and -established, and it was not intended to scrutinize them here. The point is that it is dangerous to think about MID error models as means to cover deviation from metabolic stationarity. Labeling error models are, as the authors specify, assumed to be normally distributed without bias. But whether errors introduced by metabolic instationarity are also of this type is unclear. Please clarify, to circumvent the naive conclusion that an increase in error could "heal" metabolic instationarity.

4. Reply to Response to Minor Comment 3:

Since the underlying principles of incremental model updating used in Dalman 2016 seems to be not perfectly clear, this reviewer suggests replacing or amend this cite on P3 by 10.1038/s41596-019-0204-0, where the procedure is outlined.

5. Abstract:

The notion of prediction uncertainty should be formulated more precisely, i.e. labeling prediction uncertainty, since the constraint-based modelling community often uses the term "flux prediction".

6. P4: "By quantifying ... core prediction," is hard to understand without further background.

It may be better so state why prediction uncertainty is important and validation data should be chosen wisely, instead of explaining how this is done here. Later, when the prediction and inference uncertainties are discussed, a statement is warranted what such "core predictions" are. Also, the authors may comment why MCMC is preferred over the standard PL-based approach for flux confidence bounds, whereas for prediction uncertainty the PL-based approach and not a standard MC approach is taken.

7. P13: "This theory says ... goes to infinity."

This is not generally true. An important assumption is stated later in the paragraph, but in a rather disconnected manner. Please put it in direct context. For instance, in case the data is informative and no structural non-identifiabilities are present, theory says ...

8. P13: "and the results will still hold as long as the same difference between metabolites holds also for sigma_r".

Unclear, please revise.

9. P14: "compared to methods for mechanistic modeling, where new distributions can be used for the analysis".

Unclear, please revise.

Typos:

P3: model structures that pass the Chi2 test.; the underlying errors

P6: First Chi2 (without ")

P9: can be used for; for this new method in 13C MFA; does not contain any new information (instead provide)

P13: offer a pragmatic way forward

P14: of these results is that also

P15: on errors in sigma_b

P18: Sect. 4.4 check tense

References: Check duplicates (e.g. Aitchison)

**Have the authors made all data and (if applicable) computational code underlying the findings in their manuscript fully available?**

Reviewer #2: **No: **see comments to the authors

PLOS authors have the option to publish the peer review history of their article (what does this mean?). If published, this will include your full peer review and any attached files.

Reviewer #2: No

Figure Files:

Data Requirements:

Reproducibility:

References:

---

## [Decision Letter · Decision Letter 2]

7 Mar 2022

Dear Mr Sundqvist,

We are pleased to inform you that your manuscript 'Validation-based model selection for 13C metabolic flux analysis

with uncertain measurement errors' has been provisionally accepted for publication in PLOS Computational Biology.

Best regards,

Vassily Hatzimanikatis

Associate Editor

PLOS Computational Biology

Kiran Patil

Deputy Editor

PLOS Computational Biology

Reviewer's Responses to Questions

**Comments to the Authors:**

Reviewer #2: The authors have revised their manuscript, and responded to all reviewer criticisms. I am satisfied with the clarifications to the paper.

**Have the authors made all data and (if applicable) computational code underlying the findings in their manuscript fully available?**

Reviewer #2: Yes

PLOS authors have the option to publish the peer review history of their article (what does this mean?). If published, this will include your full peer review and any attached files.

Reviewer #2: No

---

## [Editor Report · Acceptance letter]

6 Apr 2022

PCOMPBIOL-D-21-01170R2 

Validation-based model selection for 13C metabolic flux analysis
with uncertain measurement errors

Dear Dr Sundqvist,

I am pleased to inform you that your manuscript has been formally accepted for publication in PLOS Computational Biology. Your manuscript is now with our production department and you will be notified of the publication date in due course.

With kind regards,

Olena Szabo
